# A structured coalescent model reveals deep ancestral structure shared by all modern humans

Trevor Cousins, Aylwyn Scally ⓘ & Richard Durbin ⓘ ✉

Understanding the history of admixture events and population size changes leading to modern humans is central to human evolutionary genetics. Here we introduce a coalescence-based hidden Markov model, cobraa, that explicitly represents an ancestral population split and rejoin, and demonstrate its application on simulated and real data across multiple species. Using cobraa, we present evidence for an extended period of structure in the history of all modern humans, in which two ancestral populations that diverged ~1.5 million years ago came together in an admixture event ~300 thousand years ago, in a ratio of ~80:20%. Immediately after their divergence, we detect a strong bottleneck in the major ancestral population. We inferred regions of the present-day genome derived from each ancestral population, finding that material from the minority correlates strongly with distance to coding sequence, suggesting it was deleterious against the majority background. Moreover, we found a strong correlation between regions of majority ancestry and human–Neanderthal or human–Denisovan divergence, suggesting the majority population was also ancestral to those archaic humans.

Improvements in the technology to extract ancient DNA have enabled an increasingly detailed picture of human evolutionary genetics in the late Pleistocene and Holocene[1], which overwhelmingly suggests that in the last tens of thousands of years, there has been repeated separation and subsequent remixing, or admixture, of populations. Further back in time, high-coverage genomes from Neanderthals and Denisovans strongly indicate gene flow from these archaic humans into non-Africans[2–5], and more ancient gene flow from the ancestors of modern humans into the ancestors of Neanderthals[6–8]. Moreover, researchers have demonstrated that models that incorporate a contribution of ancestry within the last ~100 thousand years from an unknown archaic population better explain patterns of polymorphism in African populations than a model without such a contribution[9–16]. However, the presence of more ancient admixture events is less clear[17].

The history of effective population size changes is another important quantity in understanding evolutionary genetics[18]. The pairwise sequentially Markovian coalescent (PSMC)[19] was introduced to infer changes over time in the coalescence rate, the inverse of which can be interpreted as the history of effective population sizes. PSMC assumes that a population evolved under panmixia, with random mating in the ancestral population at all times. In light of the repeated evidence for ancestral population structure and admixture summarized above, PSMC's assumption of an unstructured evolutionary history is questionable. Moreover, theoretical analysis shows that for any panmictic model with changes in the effective population size, there necessarily exists a structured model that can generate exactly the same pairwise coalescence rate profile without changes in population sizes[20,21].

Here we address whether the use of additional information can restore identifiability. We demonstrate that the transition matrix of the PSMC hidden Markov model (HMM) has information that can distinguish a structured model from a panmictic model, even if they have matching coalescence rate profiles. We parameterize a model of ancestral population structure that leverages this information and introduce this in an HMM called cobraa. This approach can be applied to diploid

Department of Genetics, University of Cambridge, Cambridge, UK. ✉e-mail: rd109@cam.ac.uk

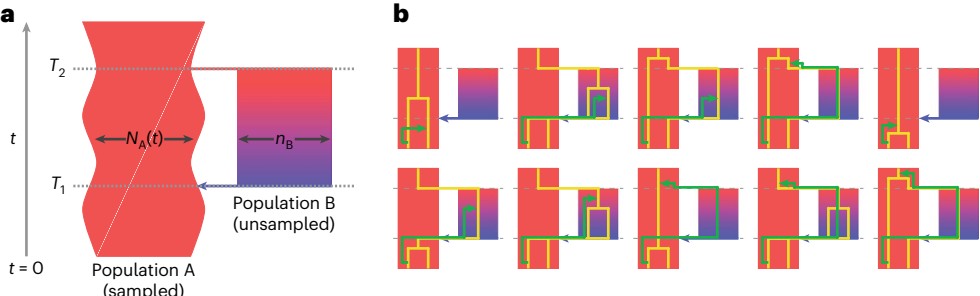

**Fig. 1 | The structured ancestry model used by cobraa. a**, Diagram of the structured model. Going forward in time, an ancestral population splits into two populations, A and B, at time $T_2$. These remain in isolation until time $T_1$ when there is an admixture event. **b**, We extend the SMC model to include the structure in **a**. We consider two sampled lineages from population A at $t = 0$.

The consequences of ancestral recombination are partitioned into ten mutually exclusive cases, with an example from each illustrated. The gold lines indicate the two chromosomes sampled from the present at a particular locus, and the green line indicates the 'floating' lineage under the SMC model, which coalesces somewhere higher up on the tree in either population A or B.

sequence data from any species, and we show a variety of different inferred histories in various mammals including humans. Applying cobraa to data from the 1000 Genomes Project (1000GP)[22–26] and the Human Genome Diversity Project (HGDP)[27,28], we show that a model of deep population structure, where modern humans are a result of two populations that diverged ~1.5 million years ago (Ma) admixing together ~300 thousand years ago (ka) in a ratio of ~80:20%, better explains the data than does a continuously panmictic model. We use posterior decoding to infer regions of the modern human genome that are derived from each population and find evidence for selection against the material from the population contributing the minority of ancestry. Moreover, we find a strong association between regions derived from the major ancestral population and human–Neanderthal or human–Denisovan divergence, suggesting that the majority population was the primary ancestral population to Neanderthals and Denisovans.

## Results

### Identifiability of structured ancestry in the SMC transitions

We consider a pulse model of population structure, where there are two populations A and B, which descend from a common, ancestral population (Fig. 1a). Looking backward in time, population A is panmictic until time $T_1$ when a fraction $\gamma$ of the lineages instantaneously derive from a new population B; A and B remain in isolation until time $T_2$ when all lineages merge into a panmictic, ancestral population. The size of population A may vary in time, but we enforce that B must be of constant size, and hence, the parameters of this model are the population sizes $N_A(t)$ in A, the population size $N_B$ in B between $T_1$ and $T_2$, the admixture fraction $\gamma$ and the split and admixture times, $T_2$ and $T_1$, respectively. To fit this model, we partitioned the ancestral recombination process into ten mutually exclusive cases according to possible migration or coalescence events, linked via a sequentially Markovian coalescent (SMC) model[29,30]. We give illustrative examples in Fig. 1b and provide the mathematical details in the Supplementary Note.

We first demonstrate that even if structured and unstructured (that is, panmictic) models have the same coalescence rate profile, they differ in their conditional distributions of neighboring coalescence times, corresponding to the transition probabilities in the SMC. We consider a structured model where A and B have a constant size and calculate its coalescence rate profile (Fig. 2a, blue line); we can then construct an unstructured model with changes in its effective population size such that it has the same rate profile[20,21] (Fig. 2a, orange line). We discretize the conditional probability distributions for the structured and unstructured model into transition matrices $Q^S$ and $Q^U$, respectively (see derivations in Supplementary Note). The relative differences between these matrices $\xi = (Q^U - Q^S)/Q^S$ are clearly nonzero, as seen in Fig. 2b. This is not a consequence of time discretization, as the

relative difference does not decrease as the number of time intervals increases (Supplementary Fig. 1). Moreover, the difference increases as the admixture fraction or duration of population separation between $A$ and $B$ increases (Supplementary Fig. 1). Thus, even if a structured and unstructured model have indistinguishable coalescence rate profiles, the conditional distribution of neighboring coalescence times provides information to discriminate between them, with likelihood differences shown in Supplementary Fig. 2.

To exploit this information and infer the parameters of our structured model, we introduce a new method for coalescence-based reconstruction of ancestral admixture, cobraa. Like PSMC, cobraa's hidden states are a set of discrete coalescence time windows, and the observations describe whether positions across the genome are homozygous or heterozygous. The emission vector describes the probability that a mutation has arisen given the coalescence time, while the transition matrix describes the probability of an ancestral recombination event changing the local coalescence time as a function of the parameters of the structured model ($N_A(t)$, $N_B$, $\gamma$, $T_2$ and $T_1$). A full description of cobraa is given in the Supplementary Note.

### Power of cobraa to infer parameters of a structured model

Using simulated data, we explored whether cobraa has power to distinguish ancestral structure from panmixia, and how well it can infer the parameters of the structured model[31]. We first tested how recoverable the admixture fraction $\gamma$ is, provided that the population sizes and split times are known. We simulated ten replicates of a 3 Gb sequence from a structured evolutionary history, where populations A and B have a constant size of $N = 16{,}000$, mutation rate per generation per base pair of $\mu = 1.25 \times 10^{-8}$ or $\mu = 1.25 \times 10^{-7}$ and recombination rate per generation per base pair of $r = 1 \times 10^{-8}$, for various combinations of the admixture fraction $\gamma$ and split/admixture times $T_1$, $T_2$. We ran cobraa until convergence (defined as the change in total log-likelihood being less than one, $\psi_{\mathcal{L}} < 1$, between consecutive expectation–maximization (EM) iterations) and plot the inferred admixture fraction in Fig. 3a. The simulated value is generally well recovered down to $\gamma$ of ~5%, although it is increasingly underestimated as it gets larger. This bias reduces as the ratio of $\mu/r$ increases.

We next simulated a structured model with $\gamma = 30\%$, over a sequence of length 3 Gb with $\mu = 1.25 \times 10^{-8}$ and $r = 1 \times 10^{-8}$, and a bottleneck from 300,000 to 40,000 years ago. Ten replicates were performed, on which we ran PSMC and cobraa until convergence. Figure 3b,c shows inference from PSMC (light blue lines) and cobraa (red lines), respectively. The purple lines indicate the simulated inverse coalescence rate (ICR), corresponding to the effective population size. PSMC detects a false peak instead of a flat population size, whereas cobraa infers changes in population A that more

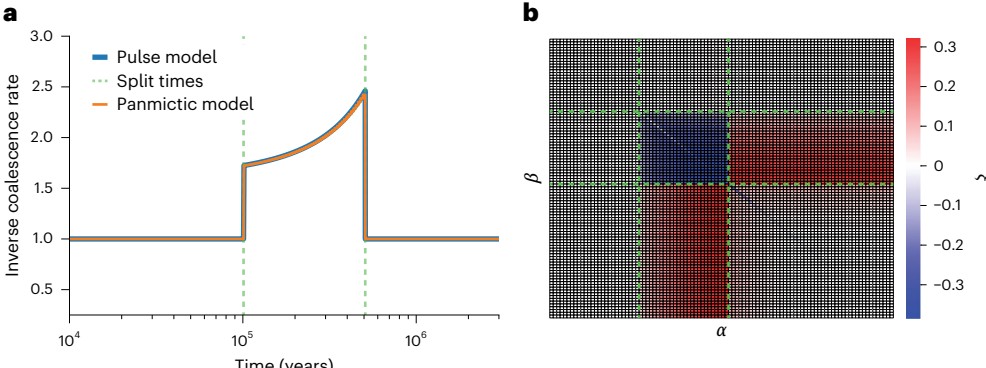

**Fig. 2 | Difference in transition matrices for matched structured and unstructured models. a**, Matching coalescence rate profiles for a structured and unstructured model. The blue line indicates the theoretical inverse coalescence rate for the structured model, where populations A and B are of constant size, the split and admixture times are given by the vertical, dashed green lines and the admixture fraction is 30%. The orange line indicates an unstructured model with size changes that generate a coalescence rate equal exactly to the structured model. **b**, A visualization of $\xi = (Q^U - Q^S)/Q^S$, the relative difference between the transition matrices for the structured and unstructured models in **a**. $Q^U$ and $Q^S$ are row stochastic matrices, where $Q(\alpha|\beta)$ describes the probability of transitioning from discretized time $\beta$ to time $\alpha$, conditional on recombination having occurred. The dashed green lines indicate the split and admixture times as in **a**.

closely reflect the simulated values. The inferred admixture fraction is relatively accurate (Fig. 3c inset). We also explored whether cobraa could recover changes in the size of population B, $n_B(t)$, as well as $n_A(t)$, and found identifiability problems (Supplementary Note), supporting our decision to maintain the size of population B constant over time.

The population size changes and admixture fraction can be inferred as part of the EM algorithm in cobraa, although the split and admixture times are considered fixed in a single run. To estimate these times, we run cobraa over various $(T_1, T_2)$ pairs, iterate each pair until convergence ($\psi_{\mathcal{L}} < 1$) and record the log-likelihood ($\mathcal{L}$). The difference between these values and the log-likelihood of the best-fitting unstructured model are shown in Fig. 3d, with the simulated pair shown in the green cell. The maximum likelihood inferred split and admixture time, shown in the yellow cell, is not exactly at the simulated value, but it is adjacent and in a relatively small neighborhood of high-scoring pairs, indicating that we have reasonable power to infer the split and admixture times. In the region of high log-likelihood values, the inferred $\gamma$ is also around the simulated value (Fig. 3e). By contrast, in the lower scoring pairs, the inferred $\gamma$ is increasingly different from the simulated value, and in the minimal $T_1$/maximal $T_2$ pairs, the inferred $\gamma$ is close to zero (Supplementary Fig. 4).

Next, we explored our ability to distinguish between a structured and a panmictic evolutionary history. We simulated from a series of paired panmictic and structured models with the same coalescence rate profiles, and calculated the difference between log-likelihoods obtained from fitting a structured and unstructured model for each dataset $\Delta_{\mathcal{L}} = \mathcal{L}_S - \mathcal{L}_U$, where $\mathcal{L}_S$ is the log-likelihood of the best-fitting structured model and $\mathcal{L}_U$ is the log-likelihood of the best-fitting unstructured model. The results for various simulated split times are shown in Fig. 3f. Consistently, we see that if the simulation was structured, then the inference from the structured model better explains the data than unstructured inference, as seen by positive log-likelihood differences, $\Delta_{\mathcal{L}}$. The differences increase with the period of separation between the split and admixture time. Conversely, if the simulation was panmictic, then structured inference is not able to explain the data any better than unstructured, as the log-likelihood differences are around 0. We further explore the power of cobraa's inference in the Supplementary Note.

### Inference on human data

We use one high-coverage, whole-genome sequence from each of the 26 distinct human populations in the 1000GP[24,26]. To see how

well an unstructured or structured model explains the data, we run PSMC (unstructured) and cobraa (structured) inference until convergence. To ensure that each sample uses the same discrete time interval boundaries, we fix the scaled mutation rate at $\theta = 0.0008$, which is close to the mean across populations (Extended Data Fig. 1a and Supplementary Table 1).

In Fig. 4, we show $N_A(t)$ as inferred from the unstructured (Fig. 4a) and structured (Fig. 4b) models on individuals from 1000GP. For cobraa, we show the composite maximum likelihood (CML) estimate of the split and admixture times across populations (Extended Data Fig. 2; Methods). The strongly positive $\Delta_{\mathcal{L}}$ values (Fig. 4c) for each population indicate that a structured model explains the data much better than a continuously panmictic model, as inferred by PSMC (model selection is discussed in Methods). The CML estimate of the split and admixture time (Fig. 4b) suggests that two populations diverged ~1.5 Ma and then subsequently admixed ~290 ka, around or shortly before the proposed origin of modern humans[32,33]. The inferred admixture fraction indicates that present-day humans derive 79% and 21% of their ancestry from ancestral populations A and B, respectively (Fig. 4c). Outside the structured period (more recently than the admixture time or more anciently than the split time), the inferred effective population changes are very similar in the two models (Fig. 4a,b). However, immediately more recently than the split time at ~1.5 Ma, in each 1000GP sample, the structured inference in cobraa infers a strong bottleneck in population A, followed by a progressive effective population size increase until the admixture time. The mean of the maximum likelihood estimates of $N_B$ is substantially larger than the size of population A (Fig. 4b, dashed blue line; further details in Extended Data Fig. 3).

We also ran cobraa on data from the Human Genome Diversity Project (HGDP)[27,28] and found very similar results (Supplementary Fig. 7) to those as inferred in 1000GP. Even in the San, who are estimated to harbor the most divergent ancestry across present-day humans[33-35], the structured model is a substantially better fit to the data than a continuously panmictic model.

When we simulate sequence data from cobraa-inferred structured models from Fig. 4b and then run unstructured (PSMC) inference, the inferred changes in population size (Supplementary Fig. 5a) are extremely similar to those inferred by PSMC in real human data (Fig. 4a), demonstrating that our structured model is compatible with previous coalescence-based estimations that assumed panmixia[36-41]. We note that our simulations suggest that the site frequency spectrum could be used to distinguish between a structured or unstructured

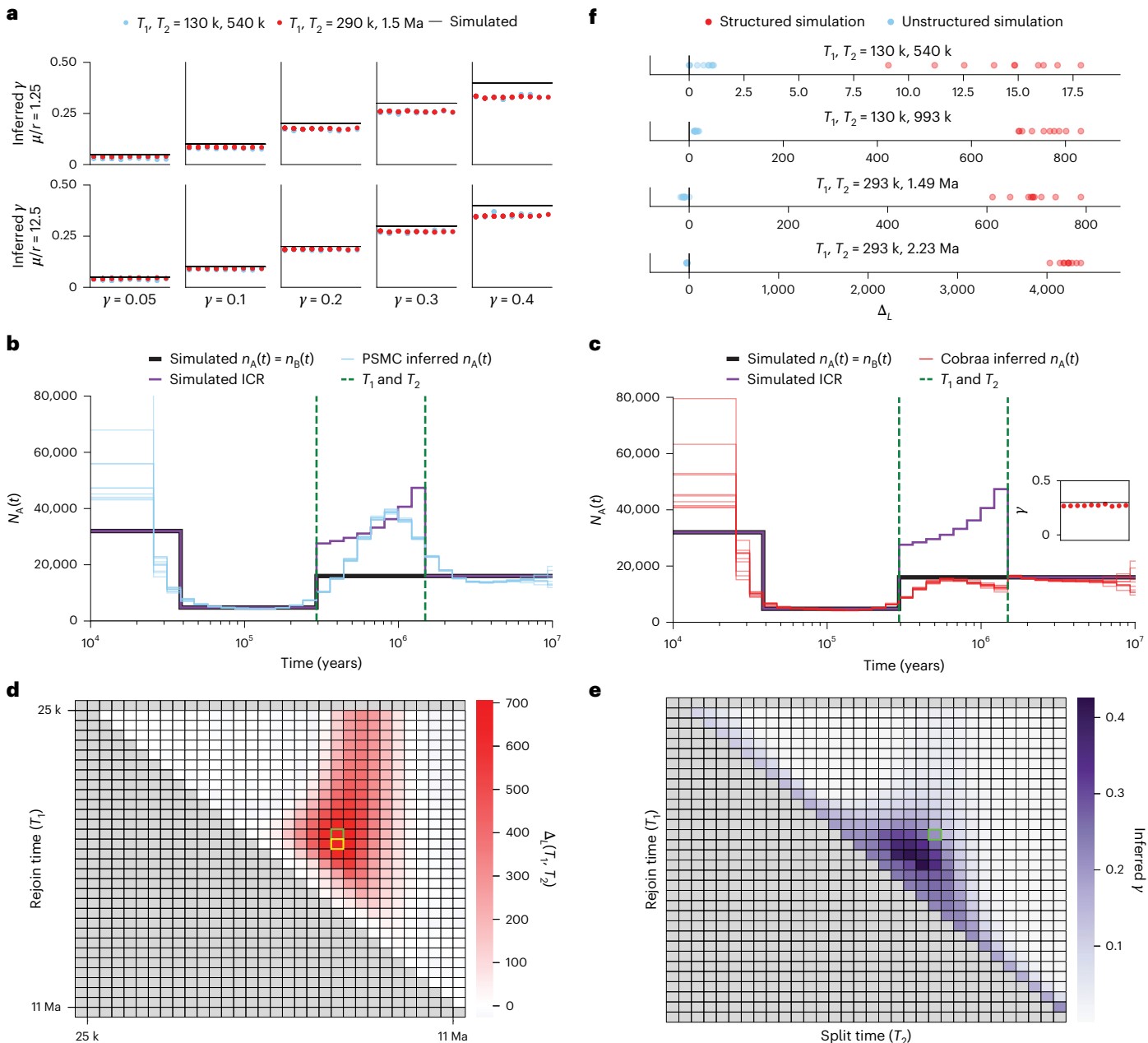

**Fig. 3 | Ability of cobraa to infer parameters of a structured model. a**, Inferring the admixture fraction $\gamma$, when the population sizes and split/admixture times are fixed at their simulated value. The simulated model has constant size in populations A and B, $\mu = 1.25 \times 10^{-8}$ (top) or $\mu = 1.25 \times 10^{-7}$ (bottom), $r = 1 \times 10^{-8}$, and 3 Gb of sequence data for various values of $\gamma$ and $(T_1, T_2)$. Ten replicates of each are shown. **b**, PSMC inference of $N_A(t)$, on a simulated structured model. The black line indicates simulated $N_A(t)$, which is the same as simulated $N_B(t)$. The green, dashed, vertical lines indicate the split and admixture times ($T_1 = 300$ ka and $T_2 = 1.5$ Ma, respectively) with $\gamma = 30\%$, $\mu = 1.25 \times 10^{-8}$, $r = 1 \times 10^{-8}$ and 3 Gb of sequence data. The purple line is the simulated ICR. **c**, Cobraa inference of $N_A(t)$ for the same structured model. The inset shows the inferred admixture fraction.

**d**, Using cobraa to search for $T_1$ and $T_2$ by iterating EM till convergence and recording the log-likelihood. The vertical axis represents the admixture time $T_1$, with values closer to the top indicating more recent times. The horizontal axis represents the split time $T_2$, with values more right indicating more ancient times. The simulated $(T_1, T_2)$ pair is highlighted in the green cell, and the maximum-likelihood $(T_1, T_2)$ pair is highlighted in yellow. **e**, Corresponding inference of $\gamma$ for each pair (relative error is shown in Supplementary Fig. 4). **f**, Difference in model fits between cobraa and PSMC, $\Delta_{\mathcal{L}} = \mathcal{L}_S - \mathcal{L}_U$, for a structured simulation (red points) and an unstructured simulation (blue points), both of which have the same coalescence rate profile. The third panel corresponds to the inference as shown in **b** and **c**. ICR, inverse coalescence rate.

model (Supplementary Fig. 6; Methods); thus, a method combining coalescence and site frequency information to infer ancestral structure could be powerful[38,42].

To explore the degree of support for the bottleneck in population A after divergence from B, we ran cobraa with various parameter constraints. We experimented with different levels of freedom in $N_B(t)$, and also tried optimization after enforcing constant size in A (Methods).

We were not able to fit a model lacking a bottleneck in A that has comparable $\mathcal{L}$, suggesting that the bottleneck is a necessary feature. We also investigated how the inference from cobraa changes in the presence of low-quality regions of the genome, with widespread linked selection, or with more natural estimations of heterozygosity, and found that none of these have a substantial effect on inference (Methods).

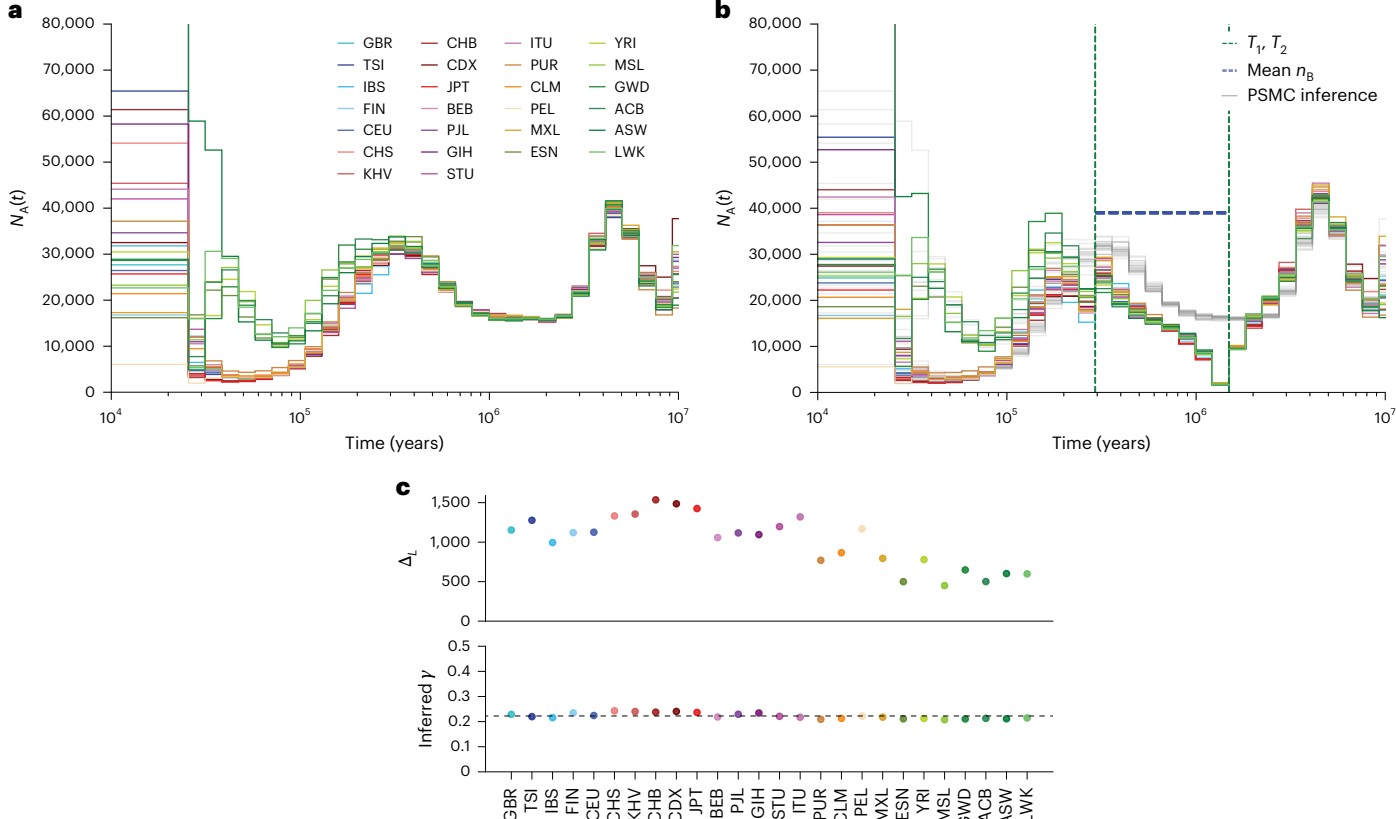

**Fig. 4 | Inference from PSMC and cobraa on 26 individuals from the 1000GP.**
**a**, PSMC's estimate of $N_A(t)$. **b**, cobraa's estimate of $N_A(t)$, with the estimated split and admixture times (~290 ka and ~1.5 Ma, respectively) shown in vertical, dashed, green lines. For direct comparison, the PSMC inference from **a** is also plotted in gray. The mean inferred size of population B, ~39,200, is shown in the horizontal, dashed, blue line. **c**, Top, the difference between the log-likelihood from cobraa's inference and PSMC's inference, $\Delta_{\mathcal{L}}$ for each population; bottom, cobraa's inferred admixture fraction $\gamma$. Identifiers for samples selected are given in Supplementary Table 1, and full names corresponding to triplet codes for the populations are given in Supplementary Table 5.

## Inferring admixed regions of the genome

We expanded the HMM of cobraa such that the hidden state represents not just the coalescence time, but also the path through the structured model taken by both lineages before they coalesce. We call this extended HMM cobraa-path. The hidden states are then a tuple $(t, c)$, where $t$ is the coalescence time and $c \in [AA, BB, AB]$ is the path choice. If coalescence occurs more recently than $T_1$ then $c = AA$; if in the structured period $T_1 \leq t < T_2$, then $c \in [AA, BB]$; if $t \geq T_2$, then $c \in [AA, BB, AB]$ (Extended Data Fig. 4). The transition and emission probabilities of cobraa-path follow naturally from cobraa and are given in the Supplementary Note. Running cobraa-path on simulations suggests that we can infer $c$ with reasonable accuracy using posterior decoding (Extended Data Fig. 5).

Using the inferred structured model from the previous section, we decoded each of the 26 samples from 1000GP and analyzed the marginal posterior probability of each lineage path at each position, $P(c_i | X)$. We then condition these estimates on the coalescence time being larger than the admixture event, $P(c_i | X, t > T_1)$, to reduce confounding. The correlation between $P(c_i \in [AA, AB, BB] | X, t > T_1)$ and the distance to closest coding sequence (dcCDS) is shown in Extended Data Fig. 6a. There is a weak but significant Spearman correlation between dcCDS and both $P(c_i = AB | X)$ and $P(c_i = BB | X)$ (-0.078 and -0.075, respectively, with $P < 2 \times 10^{-3}$ for all populations (Methods; Supplementary Table 2)), suggesting that genetic material from the minority population was selected against in modern humans post admixture. To confirm this, we examined the correlation between $P(c_i | X, t > T_1)$ and a high-resolution B map[43], which integrates functional and genetic map information to estimate the strength of background selection across the genome. The Spearman correlation between $P(c_i \in [AB, BB] | X, t > T_1)$ and the B-map is

larger than that with dcCDS and also significant (-0.228 and -0.301 for AB and BB, respectively, with $P < 2 \times 10^{-3}$ for all populations; Extended Data Fig. 6b and Supplementary Table 2), supporting the suggestion of negative selection on material from population B.

We next looked at regions of the genome that are enriched or depleted for inferred B ancestry. To do this, we define a test statistic $H(x)$, which calculates for each position $x$ the expected amount of admixture across all 1000GP samples (Methods). For 1-kb regions that are in the top or bottom 1% of $H(x)$ values, we looked to see whether they overlap with any protein-coding genes. We found 680 protein-coding genes that overlap with regions in the top 1% of $H(x)$ (~1.06-fold enrichment relative to neutral expectation, $P = 1 \times 10^{-17}$; Methods), which we call admixture-abundant genes (AAGs), and 1,287 protein-coding genes that overlap with regions in the bottom 1% of $H(x)$ (~1.92-fold enrichment relative to neutral expectation, $P < 1 \times 10^{-100}$), which we call admixture-scarce genes (ASGs). We show an example of $H(x)$ and inferred the probability of admixture at *KNG1* (an AAG) and *FOXP2* (an ASG) in Supplementary Fig. 8.

To examine whether AAGs or ASGs are associated with any biological processes, we performed gene ontology (GO) analysis[44,45] using PANTHER[46] (Methods). The 680 AAGs showed an 11.6-fold enrichment for genes associated with neuron cell–cell adhesion, 8.5-fold enrichment for startle response, 6.2-fold enrichment for neuron recognition, 3.7-fold enrichment for neurotransmitter transport, 2.5-fold enrichment for chemical synaptic transmission and 2.2-fold enrichment for circulatory system processing. Additionally, a twofold depletion in genes associated with gene expression was reported. Similarly, numerous associations between the 1,287 ASGs and biological processes were

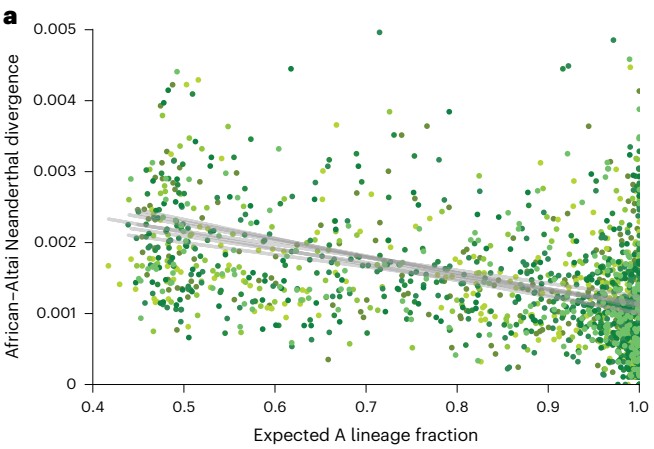
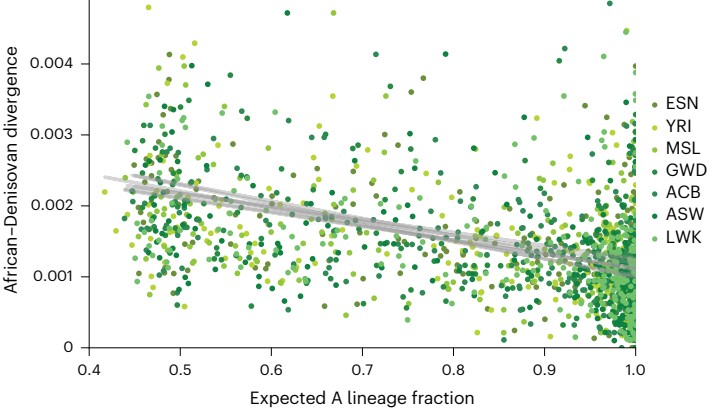

**Fig. 5 | Relationship between human–archaic divergence and cobraa-path's expected fraction of ancestry from the A lineage. a**, Divergence to the Altai Neanderthal sequence, plotted against the mean of $P(c = AA) + P(c = AB)/2$, calculated in windows of 10 kb and shown as points (subsampled for clarity) for each of seven African individuals. **b**, Corresponding plot for divergence

to the Denisovan sequence. As confidence in a region being assigned to the A lineage increases, both human–Neanderthal and human–Denisovan divergence decreases. This suggests that Neanderthals and Denisovans derive from population A. The gray lines show a linear line of best fit for each population.

reported. Notably, it reported an 8.3-fold enrichment for genes associated with pre-miRNA processing, fourfold enrichment for cortical actin cytoskeleton organization and 3.6-fold enrichment for Golgi-to-plasma membrane transport, among many others. Moreover, ASGs were found to have a 3.7-fold depletion in genes associated with adaptive immune response, 7.7-fold depletion in lymphocyte-mediated immunity, 25-fold depletion in detection of chemical stimulus involved in sensory perception of smell and >100-fold depletion in antimicrobial humoral response. More subprocess information is given in Supplementary Tables 3 and 4, with full GO analysis (including parent categories) available online.

We next investigated whether population A or B is closer to Neanderthals and Denisovans. To test this, we took 10-kb windows every 10 Mb and plotted the mean sequence divergence to diploid individuals in these regions against the fraction of the region expected by cobraa-path to come from the A lineage (the mean of $P(c = AA) + P(c = AB)/2$; Fig. 5). We did this for representative individuals of African ancestry to avoid issues with post-out-of-Africa archaic admixture[2–4]. For regions confidently assigned to A (Fig. 5a,b, right), the divergence is significantly lower than regions assigned to both A and B (Fig. 5a,b, left; Spearman correlations −0.4 to −0.53, $P < 1 \times 10^{-10}$ for all correlations). No regions were assigned wholly to population B because we have low power to identify homozygous BB segments, which have a prior probability of 0.04 ($\gamma = 0.2$ squared; Extended Data Fig. 7). The plots for Neanderthal and Denisovan are essentially identical. These results suggest that the average divergence to the archaics from the A and B lineage is ~0.0012 and ~0.0028, respectively, consistent with population A rather than B being ancestral to the archaics. Most coalescence in the A lineage will be more recent than its founding bottleneck, whereas coalescence in B would be more ancient as its effective population size was larger between $T_1$ and $T_2$.

### Application to other species

We next ran cobraa on other species, first considering two species with recent high-quality diploid assemblies for which 95% of the genome is callable. The parti-colored bat (*Vespertilio murinus*)[47] showed little evidence for structure using cobraa, with a single $T_2$ value showing positive but relatively low $\Delta_{\mathcal{L}}$, with $\gamma$ near 0 (Extended Data Fig. 8a–c). For the common dolphin (*Delphinus delphis*), cobraa did find evidence for structure ($\Delta_{\mathcal{L}} \simeq 4000$), with $(T_1, T_2) \approx (100$ ka, 650 ka$)$ in a similar period of the coalescent history as the best-fitting model in humans,

and $\gamma = 0.22$ (Extended Data Fig. 8d–f). However, the actual $N_A(t)$ histories differ from those for humans (Extended Data Fig. 8f).

We also applied cobraa to chimpanzees and gorillas, using short-read whole-genome sequence data[48] aligned to the human reference genome (Extended Data Fig. 8g–l). In the eastern lowland gorilla (*Gorilla beringei graueri*), a structured model with $(T_1, T_2) \approx (15$ ka, 150 ka$)$ and $\gamma = 0.29$ fits the data better than an unstructured model ($\Delta_{\mathcal{L}} = 900$; Extended Data Fig. 8g,h). Splits between eastern and western gorillas had previously been inferred around 150 ka, together with indications of more recent contact[49], consistent with this inferred period of structure. The cobraa model did not detect the 3% deep ghost admixture recently reported[50].

For the Nigeria–Cameroon chimpanzee (*Pan troglodytes ellioti*), inference from cobraa suggests a very recent (possibly ongoing) structure, with a high likelihood model ($\Delta_{\mathcal{L}} = 718$) suggesting $(T_1, T_2) \approx (2.5$ ka, 83 ka$)$ and $\gamma = 0.49$ (Extended Data Fig. 8j,k). We note that both the admixture time and fraction are at their boundaries, suggesting that the cobraa model is not well-suited to this dataset. Perhaps this reflects continuing gene flow between different chimpanzee subspecies[51].

In addition, for both gorilla and chimpanzee, there is a streak on the right-hand side of Extended Data Fig. 8g,j, corresponding to maximal $T_2$ with very low $\gamma$. We suggest this may be artefactual, arising from alignment of short reads to a human reference (only 49% of the genome was assessed as callable using our filters; Methods).

We conclude that with different species, cobraa makes different inferences, in some cases providing little or no evidence for ancestral structure, whereas in other cases identifying plausible candidate periods of ancestral structure.

### Discussion

We introduce here a coalescent-based method to infer a structured ancestry from a diploid genome sequence, which we have used to infer a deep split in human ancestry ~1.5 Ma, rejoining ~300 ka, around the time of the earliest anatomically modern human fossils[32]. The evidence that Neanderthal and Denisovan genomes diverged more recently from ancestry on the major (A) lineage, rather than the minor (B) lineage, supports the conclusion that the structure we infer is neither artefactual nor arbitrary, as the inference procedure is independent of these archaic genomes. A simplified diagram of human evolutionary history, with cobraa's contribution highlighted in red, is shown in Fig. 6. We provide evidence of general selection against introgressed material from the minor (B) lineage, while also seeing enrichment of

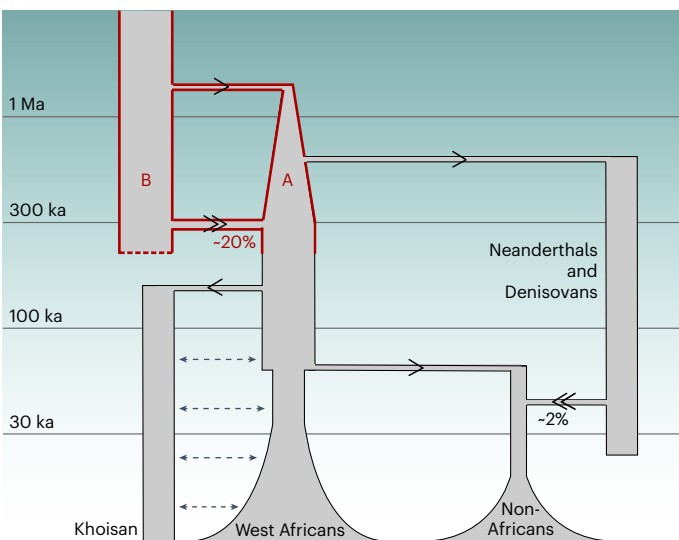

**Fig. 6 | A simplified model of human demographic history, as inferred by cobraa.** A simplified model of human demographic history showing deep population structure ~1.5 Ma to ~300 ka ago shared by all present-day humans, as inferred by cobraa (red). Arrows indicate the direction of gene flow, with admixture events (double arrows) labeled by their percentage genetic contribution to the recipient population. Of the two ancestral branches A and B, A represents 80% of subsequent ancestry and features a sharp bottleneck immediately after its founding. Dashed arrows between Khoisan and other African populations reflect the fact that this divergence, the deepest among present-day human populations, has involved ongoing or intermittent gene flow[33,37,42]. The y axis represents time in years before the present.

introgression specifically in a set of categories associated with neuronal development and processing. The admixture percentage of ~20% is much higher than the fraction of Neanderthal or Denisovan admixture into present-day non-African populations, but was not discernible using standard f-statistics because it is shared by all present-day humans. We note that similarly high admixture fractions with enrichment of admixture in gene categories important in speciation have been seen at the base of speciation/radiation events in other taxa, for example, in Lake Malawi cichlids[52].

Recently, ref. 16 also proposed deep ancestral structure in the ancestors of modern humans. Their best-fitting model suggested an initial divergence ~1.7 Ma into two populations, 'stem 1' and 'stem 2', with continuous gene flow between them until ~500 ka, at which point, stem 1 split into populations—1S ancestral to Khoisan and 1E ancestral to all other modern human subpopulations. Stem 2 is estimated to have contributed 70% of ancestry into the Khoisan lineage ~120 ka, and 50% of ancestry into stem 1E ~100 ka. The initial split time of ~1.7 Ma is similar to ~1.5 Ma estimated by cobraa. In contrast, cobraa estimates a single admixture event ~300 ka, substantially more ancient than the earliest admixture reported in ref. 16. The differences could be due to the space of models. For example, cobraa requires populations A and B to be in isolation after they split, contrasting with the inferred continuous gene flow between stem 1 and 2 in ref. 16. Conversely, ref. 16 assumes a constant population size in stems 1 and 2, whereas cobraa found that an early bottleneck in population *A* was important when fitting to human data.

Numerous authors have reported evidence for there being more recent contributions of unknown archaic ancestry to modern humans, especially in West Africans[9–16,53,54]. Parametric estimates vary, although all models of structure in West Africans infer that admixture occurred more recently than ~150 ka[13,14], with some inferring it more recently than 50 ka[9–12,15]. Moreover, the inferred population divergence time is always estimated as being more recent than 1 Ma. Although this

appears to be a different event to the one that we describe, not shared by all present-day humans, these inferences suggest a plausible reason why the cobraa-inferred maximum likelihood estimates of the split and admixture time in West Africans are more recent than the CML estimate (Extended Data Fig. 2).

Technically, we have demonstrated that the conditional distribution of adjacent coalescence times has information about ancestral structure, partially overcoming the pairwise coalescence rate identifiability problem[20,21]. This approach can be applied to any diploid genome sequence, and we have shown that a variety of different past admixture events are inferred by cobraa in different species' genomes, including failing to identify any significant event in the parti-colored bat. However, there are clearly limitations to the sensitivity of the approach and the complexity of past structure that can be inferred. Recent theoretical work demonstrates that the joint density of the first and second coalescence events from three lineages can also distinguish population structure from population size changes[55]. Thus, a method estimating the second as well as the first coalescence rate could help extend our approach.

There are several caveats to our approach. We have shown that a pulse model of structure better fits the human data than does a continuously panmictic model, but even if we limit ourselves to older events shared by all modern humans, our evolutionary history is very likely more complex than this[16]. We can not rule out multiple split and admixture events, or more continuous gene flow. Additionally, our assumptions of an absence of selection and constant mutation and recombination rates across the genome are false[56–69]. However, previous studies that incorporated realistic variation in mutation and recombination rates into simulations have shown that they only have negligible consequences for SMC-based inference[19], and[70] shows that genomic variation in linked selection does not have a noticeable effect on the identification of structure.

The model of ancestral structure we propose raises intriguing questions about the relationship of lineages A and B to previously identified hominins. Archeological evidence suggests numerous forms of archaic hominins with it unclear which, if any, contributed directly to the ancestry of modern humans[17]. Various *Homo erectus* and *Homo heidelbergensis* populations that are potential candidates for lineages A and B existed both in Africa and elsewhere in the relevant period. It is tempting to ascribe the sharp bottleneck that we infer in lineage A after separation from lineage B to a founder event potentially involved with migration and physical separation. Furthermore, the ancestors of Neanderthals and Denisovans were in Eurasia before modern humans expanded there, and we can ask whether the gene flow from the ancestors of modern humans into Neanderthals[6–8,71,72] came from A or B, and also how the proposed archaic gene flow event into Denisovans[2,4] was related to these populations. Unfortunately, we were not able to run cobraa on currently available Neanderthal and Denisovan datasets (Methods). Further clarifying the genetic contributions to modern humans and connecting them to the fossil record is an ongoing challenge.

## Online content

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

## Methods

No specific ethical approval was required for this study.

### Model summaries

Cobraa is an HMM that builds on the PSMC framework. The hidden states are the discretized coalescence times across the genome, and the observations are the series of homozygotes or heterozygotes in a diploid genome sequence. The model parameters are $N_A(t)$, $N_B(t)$, $\gamma$, $T_1$ and $T_2$, which are, respectively, the population sizes in the sampled population (A), the population sizes in the ghost population (B), the admixture fraction, the admixture time and the split time.

The emissions describe the probability of a mutation arising given a particular coalescence time. The transition matrix $Q(N_A(t), N_B(t), \gamma, T_1, T_2)$ is governed by the SMC framework[29,30], as a function of a structured model's parameters. Calculating the transition probabilities required considering the ten distinct possibilities for changes in coalescence time due to recombination or migration, as shown in Fig. 1b. The population sizes and the admixture fraction can be optimized as part of the EM algorithm, although the split/admixture times are fixed and are searched through independent model runs. We also note that when the admixture fraction $\gamma$ is equal to zero, then this corresponds to an unstructured model exactly as in PSMC; therefore, PSMC is nested in cobraa.

When running cobraa on the 1000GP data, we enforce every sample has the same discrete time interval boundaries by fixing $\theta$ across all populations (the time discretization scheme is given in the Supplementary Note). This is so that we can use a CML search over all possible pairings of the split and admixture time. More explicitly, for a population $j$ and split and admixture time of $t_1$ and $t_2$, we get the log-likelihood $\mathcal{L}(j, t_1, t_2)$ by running cobraa till convergence. We then select the CML time pairing with $\mathcal{C}(t_1, t_2) = \arg\max_{t_1, t_2} \sum_j \mathcal{L}(j, t_1, t_2)$. We enforced that $N_B(t) = k$ for all $t$, where $k$ was optimized as part of the EM algorithm, due to identifiability problems between $N_A(t)$ and $N_B(t)$ (Supplementary Figs. 23 and 24). Similar to PSMC, we can take advantage of the rarity of recombination events to increase cobraa's computational speed (and decrease memory usage) $b$-fold by binning the genome into windows of $b$ base pairs. Parameter inference is not substantially affected with values of $b$ up to 100, as seen in Supplementary Fig. 3, so from here, we set $b = 100$.

We expanded cobraa into a second HMM, cobraa-path, whose hidden states are the ancestral lineage path, $c$, and the discretized coalescence times. If the sampled lineages coalesce more recently than the admixture event, then they can only coalesce in population A, so $c = AA$. If they coalesce more anciently than the admixture time but more recently than the split time, then either both lineages stayed in population A or they both migrated to B so $c \in [AA, BB]$. If they coalesce more anciently than the split time, then they either both stayed in A, both migrated to B, or one stayed and one migrated, so $c \in [AA, AB, BB]$ (Extended Data Fig. 4). The emission probabilities follow from cobraa, although are repeated across different values of $c$. For example, given a coalescence of $t$ that is more ancient than population divergence, the probability of observing a mutation is the same for each $c \in AA$, BB, AB. The transition probabilities also follow naturally from cobraa, although do not require explicit calculations of the lineage path at the previous locus. We also note that if $\gamma = 0$, then cobraa-path reduces to standard PSMC.

The main advantage of using cobraa-path is that we can decode the HMM to infer the regions of admixture, that is, where the ancestral lineage path went partially or wholly through population B. Using the forward/backward algorithm, we can thus rapidly obtain the joint posterior probability of the ancestral lineage path and coalescence time at each position $P(c_i, t_i | X, \Theta)$, where $X$ is the observed data and $\Theta$ is the set evolutionary parameters given to cobraa-path. The marginal posterior probabilities of each ancestral lineage path $c$ are easily obtained by summing over the coalescence times $P(c_i | X, \Theta) = \sum_\tau P(c_i, t_i = \tau | X, \Theta)$.

To calculate the expected amount of admixture at each position $x$, we define:

$$H(x) = \frac{\sum_j P\left(c_x^j = BB, t_x^j > T_1 | X^j\right) + (1/2)P\left(c_x^j = AB, t_x^j > T_1 | X^j\right)}{\sum_j P\left(t_x^j > T_1 | X^j\right)}$$

where $j$ represents each population, $X^j$ is all observed data for population $j$ and $c_x^j$ and $t_x^j$ are the ancestral lineage path and coalescence time, respectively, for population $j$ at position $x$.

Full mathematical details of cobraa and cobraa-path are given in the Supplementary Note.

### Associations between cobraa-path and functional information

The positions of genes and their annotations were obtained from HAVANA, as downloaded from GENCODE. We used the B-map as inferred in the YRI in ref. 43 and lifted over from GRCh37 to GRCh38 (ref. 73). We believe that using the YRI B-map for all populations is sufficient because ref. 43 reports that B-maps inferred in different populations all have a Pearson correlation of >0.999 with the YRI B-map. Positions that did not pass the GRCh38 mappability mask were excluded from the analysis. To account for chromosomal linkage when calculating the significance of correlation between the probability of an admixed region and dcCDS or B-value (Supplementary Table 2 and Extended Data Fig. 6), we only used genomic positions that were 100 kb or more apart.

$H(x)$ was inferred every 1 kb to save disk space, and positions not passing the GRCh38 mappability mask were excluded from analysis. This left 2,158,664 positions, at which $H(x)$ could be confidently calculated, and thus 21,587 positions in the top or bottom 1% of $H(x)$ values. Forty-eight percent of the 2,158,664 callable positions occur within a protein-coding gene (the start and stop positions of each gene were taken from HAVANA). Using a binomial distribution with $n = 21,587$ and $P = 0.48$, we thus would expect the top or bottom 1% of $H(x)$ to hit protein-coding genes $np = 10,362$ times. The top 1% of $H(x)$ hit a protein-coding gene 10,991 times, which is a 1.06-enrichment relative to a neutral expectation ($P = 1.08 \times 10^{-17}$ from a two-sided Binomial test with $n = 21,587$ and $P = 0.48$). These hits occurred in 680 distinct protein-coding genes, which we call the AAGs. The bottom 1% of $H(x)$ hit a protein-coding gene 19,974 times, which is a 1.92-fold enrichment relative to a neutral expectation ($P < 1 \times 10^{-100}$ from a two-sided Binomial test with $n = 21,587$ and $P = 0.48$). These hits occurred in 1,287 distinct protein-coding genes (ASGs).

For the GO analysis, we entered our AAGs or ASGs into geneontology.org and only considered associations where $P$ value (two-sided Fisher's exact test) and FDR (Benjamini–Hochberg) <0.05. In Supplementary Tables 3 and 4, superclasses of each process are not shown, although are available to download from Zenodo[74].

### Processing 1000GP and HGDP data

We took high-coverage whole-genome-sequence cram files for one individual in each of the 26 populations from the 1000GP. These are aligned to GRCh38. The cram files were converted to bam and indexed with samtools[75,76]. The genotype likelihoods were calculated with bcftools mpileup[77] by skipping alignments with mapping quality less than 20, skipping bases with base alignment quality less than 20 and setting the coefficient for downgrading mapping quality to 50. SNPs were called using bcftools and all indels were excluded. Variants were then designated as uncallable if the minimum mapping quality was less than 20, the minimum consensus quality was less than 20 or the coverage was less than half or more than double the mean coverage. Finally, we designated all regions in the strict mappability mask for GRCh38 as uncallable. Uncallable positions are labeled as missing data in the HMM. See Supplementary Table 1 for information regarding the number of heterozygous, homozygous and uncalled positions for each individual.

The triplet codes used by the 1000GP for each population is given in Supplementary Table 5.

HGDP data were processed in exactly the same way. All human plots assume a generation time of 29 years[78] and a mutation rate per generation per base pair of $1.25 \times 10^{-8}$ (refs. 62,79).

## Processing data for other species
We downloaded the processed variants for the dolphin and bat from Sanger's Genome After Party portal (accession codes GCA_949987515.1 and GCA_963924515.1, respectively). The Hi-Fi PacBio long reads were mapped back to the reference genome with minimap2 (ref. 80), then variants were called with deepvariant[81]. We then masked sites if they had less than half or more than double the mean coverage (33.8 and 38.9 for dolphin and bat, respectively), or if the conditional genotype quality was less than 50. After filtering, more than 95% of the genome was designated as callable. For the dolphin, we used a per generation per base pair mutation rate of $2.56 \times 10^{-8}$ and generation time of 21.1 years[82,83]. We used a generation time of 2 years for the bat and assumed a mutation rate per generation per base pair of $1 \times 10^{-8}$.

We downloaded processed primate data from ref. 48. We took the VCF files and masked positions according to the given bed files, which described sites where coverage was less than five and regions that did not pass the quality filters as discussed in their paper. After filtering, only 49% of the genome was designated as callable. For the eastern lowland gorilla, we arbitrarily chose the individual labeled 'Mkubwa', and for Nigeria–Cameroon chimpanzee, we chose the individual labeled 'Akwaya Jean'. We note that the analysis on other individuals looked similar. For the gorilla, we used a mutation rate per base pair per generation of $1.43 \times 10^{-8}$, and a generation time of 19 years; for the chimpanzee, a mutation rate per base pair per generation of $1.78 \times 10^{-8}$ and a generation time of 24 years were used[84].

## Model convergence and fitting
For structured and unstructured model fitting, we iterated the EM algorithm until the change in log-likelihood was less than 1. The value of 1 was somewhat arbitrary, but is a convenient stopping criterion that allowed us to be consistent across different models.

To penalize the fit of the structured model due to it having more parameters, we calculate the AIC and compare this to the AIC for the unstructured model. The number of parameters in the unstructured model is 33 (32 population size change parameters and 1 recombination rate parameter), and the number of parameters in the structured model is 37 (32 for the population size changes in population A, 1 for the recombination rate, 1 for the admixture fraction, 1 for the size of population B and 2 for the split and admixture times). We show the AIC in Supplementary Table 6. Due to the AIC being substantially lower in the structured model, we conclude that the difference in model likelihoods is not because of the difference in the number of parameters.

To check that we were not overfitting, we took the inferred parameters on seen data (training) and calculated the log-likelihood of unseen data (testing), by taking a new individual from each population. As shown in Supplementary Table 7, the differences in log-likelihood for the structured and unstructured model in the test data are also strongly positive, suggesting that a better-structured model is not due to overfitting.

## Effect of different parameter constraints
To investigate whether the inferred bottleneck (Fig. 4b and Supplementary Fig. 7) is attributable to population A, we reran cobraa after relaxing the constraint that $N_B$ must be constant (Supplementary Fig. 9). Supplementary Fig. 9a indicates that cobraa still infers a bottleneck in $N_A(t)$, with estimates of $N_B(t)$ generally being large immediately post divergence and decreasing until the time of admixture, though with greater variance across populations compared to $N_A(t)$. The inferred admixture fraction is extremely similar, and the fit of this model is

slightly better than when we enforced constant $N_B$ (Supplementary Fig. 9b), which is unsurprising due to there being more parameters. We also examined how well the model fits the data with $N_B(t)$ not being large post divergence, and found that this model is still well supported (Supplementary Fig. 9b,c).

To explore the degree of support for the bottleneck inferred by cobraa soon after the split at ~1.5 Ma, we constrained the parameters to search for a constant $n_A(t)$ during the structured period. Removing the bottleneck in this fashion is not well supported by the data, as seen in Supplementary Fig. 10. Notably, the likelihood difference $\Delta_{\mathcal{L}}$ is often negative, and the split/admixture times and admixture fraction are not consistent, indicating that even a panmictic model fits the data better than this constrained structured model, and that the variation in $n_A(t)$ is necessary for the better cobraa fit.

To search for the optimal split and admixture time, we ran cobraa over various values of $T_1$ and $T_2$, independently for each population representative. In each case, we iterated till convergence ($\psi_{\mathcal{L}} < 1$) and recorded the $\mathcal{L}$. The differences in $\mathcal{L}$ between each pair and the panmictic inference from PSMC ($\Delta_{\mathcal{L}}(\hat{T_1}, \hat{T_2}) = \mathcal{L}_S(\hat{T_1}, \hat{T_2}) - \mathcal{L}_U$) are shown in Extended Data Fig. 2a, where red indicates positive $\Delta_{\mathcal{L}}(\hat{T_1}, \hat{T_2})$, blue indicates negative and white indicates zero. The maximum varies a bit between samples, but is always within one or two cells of the CML except for West African populations, where a more recent structured event is preferred.

## Effect of long stretches of missing data on HMM inference
Low-confidence regions of the genome are labeled as missing data in the observations of the HMM. To check that this was not artificially inflating likelihood differences between the panmictic or structured model, or biasing inference, we reran inference after removing the centromeres and telomeres, which are the longest stretches of sequence marked as uncallable by the GRCh38 mappability mask. This involved splitting each chromosome into two parts, the first of which begins after the end of the telomere and ends before the start of the centromere. The second part starts after the end of the centromere and ends before the start of the telomere. Doing this for each chromosome resulted on average in removing ~100 Mb of missing data in each population. These new sequences were then given to PSMC or cobraa to run inference with composite-likelihood optimization.

The resulting inference was practically indistinguishable from the full data (Supplementary Fig. 11). The differences in $\Delta_{\mathcal{L}}$ and the inferred admixture fraction between each dataset were extremely similar, as seen in Supplementary Fig. 11, suggesting that the evidence for the structured model is not due to large regions of missing data in the HMM.

## Effect of fixing θ across populations
To enforce parameter estimates across populations using the same discrete time interval boundaries, we fixed the scaled mutation rate $\theta$ at 0.0008, despite there being as much as a ~45% difference between the highest and lowest population (PEL = 0.00069 and ESN = 0.0010; Extended Data Fig. 1a and Supplementary Table 1). We checked that our inference was not misled by this constraint by rerunning the analysis with $\theta$ inferred in the natural sense (Supplementary Fig. 12). The inferred split and admixture times are noisier (Supplementary Fig. 12b) due to the time interval boundaries not aligning, but the inferred population sizes, admixture fraction and likelihood differences are similar. Thus, we conclude the evidence for the structured model is not due to fixing $\theta$.

## Effect of widespread linked selection
It has been demonstrated that widespread linked selection is pervasive in humans[43,85], although PSMC and cobraa assume the genome evolves neutrally. Wrongly assuming neutrality has been shown to affect demographic inference[86–88], although solutions have been proposed[70,89–91]. To check that widespread linked selection is not falsely interpreted as

structure, we ran cobraa on an unstructured evolutionary history with linked selection. We used the SLiM[92] simulated data in ref. [70], where the coalescent rate profile was constructed to mimic that as inferred in West Africans, and the distribution of fitness effects was chosen according to parameters that were inferred in humans[93]. Further, $N$ was scaled down to avoid excessive memory usage, and $\mu$ and $r$ were chosen such that diversity as a function of distance to exon imitated that as observed in humans. For inference, we ran PSMC and cobraa until the change in log-likelihood in subsequent iterations of the EM algorithm was less than 0.1. In Supplementary Fig. 13, we show that cobraa is not able to explain the data any better than PSMC, indicating that widespread selection is not interpreted as structure.

## Site frequency spectrum

In addition to coalescence-based approaches, it is possible to infer population history from the site frequency spectrum (SFS)[94–99]. However, even for inference of a panmictic history (as assumed by most methods), this is an ill-posed problem[100–104], in that many different size histories can generate the same SFS. Despite this, we note that our simulations suggest the SFS for a structured model is distinct from the SFS for an unstructured model with the same coalescence rate profile, as shown in Supplementary Fig. 6. This contrasts with the identifiability problem from the pairwise coalescence rate profile[20,21], and in principle suggests that a method that uses the SFS to jointly estimate population size changes and ghost admixture could have power to detect the structured event we propose[99].

## Processing Neanderthal and Denisovan genomes

We downloaded the high-coverage Altai Neanderthal and Denisovan variants from http://ftp.eva.mpg.de. We polarized the ancestral allele by ensuring that the human reference allele matched the chimpanzee and gorilla allele, and excluding all sites that did not satisfy this. Variant positions were aligned to GRCh37, so we used LiftOver to convert these to GRChg38 (ref. [73]). To adjust for phasing uncertainty in both humans and archaics, we randomly sampled the genotypes (assigning heterozygous sites with probability half to each of the possible bases). We calculate divergence between the focal individual and the archaic in windows of 10 kb, noting that the signal for each was not overly different, and took cobraa-path's mean probability of admixture from A or B in windows of the same size.

As a prelude to running cobraa, we attempted to reproduce a PSMC analysis on the same archaic genomes[2,5]. Despite using the same filters as reported (we filtered out sites according to the masks uploaded to http://ftp.eva.mpg.de/)—and also experimenting with our own depth and quality filters—we were unable to get an $n_A(t)$ curve that ever joined with the modern human curve. In particular, the inferred population size was extremely large more anciently than ~1 Ma, suggesting regions of false heterozygosity that would make cobraa inference meaningless.

## Statistics and reproducibility

For the 26 populations in the 1000GP, we chose one sample per population as this is all cobraa requires. We chose the first sample listed per population. No data were excluded from the analyses. The code used to perform analysis on real data and simulations is available at github.com/trevorcousins/cobraa/reproducibility.

## Reporting summary

Further information on research design is available in the Nature Portfolio Reporting Summary linked to this article.

## Data availability

The 1000 Genomes Project and HGDP-aligned sequence data sets that we used are available at internationalgenome.org/data-portal/data-collection/30x-grch38 and internationalgenome.org/data-portal/data-collection/hgdp, respectively. The GRCh38 mappability mask was downloaded from ftp.1000genomes.ebi.ac.uk/vol1/ftp/data_collections/1000_genomes_project/working/20160622_genome_mask_GRCh38. The processed great ape data was downloaded from eichlerlab.gs.washington.edu/greatape/data/. The bat and dolphin genome were downloaded from the Sanger Genome After Party Portal gap.cog.sanger.ac.uk. Processed 1000GP sequence data, summaries of posterior decoding, the list of AAGs and ASGs, and full GO analysis are available to download and analyze at Zenodo[74].

We downloaded the GENCODE annotations from ftp.ebi.ac.uk/pub/databases/gencode/Gencode_human/release_45/gencode.v45.chr_patch_hapl_scaff.basic.annotation.gff3.gz, and performed gene ontology analysis at geneontology.org.

## Code availability

cobraa is freely available to use and download at github.com/trevorcousins/cobraa (ref. [105]). The code was written in Python 3 and was optimized using numba[106]. We used the PSMC algorithm as built into the cobraa codebase for all unstructured analysis. Key scripts for reproducibility are also given at github.com/trevorcousins/cobraa/reproducibility.

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

## Acknowledgements

We are grateful to members of the Durbin and Scally groups, especially R. Schweiger and C. Siu, for helpful discussion. We are also grateful to D. Reich for his comments on an early version of the manuscript, and to the Tree of Life Programme at the Wellcome Sanger Institute for making the *D. delphis* genome available prepublication together with heterozygous variant calls for it and *V. murinus*. T.C. was funded by a Wellcome Postgraduate Studentship (108864/B/15/Z), and R.D. by Wellcome Investigator Award (207492/Z/17/Z). The funders had no role in study design, data collection and analysis, decision to publish or preparation of the manuscript. For the purpose of open access, the author has applied a CC BY public copyright license to any author-accepted manuscript version arising from this submission.

## Author contributions

T.C. developed the theory and the software, carried out the experiments and wrote the manuscript draft, with supervision from A.S. and R.D. All authors conceived the project, and edited and approved the manuscript.

## Competing interests

The authors declare no competing interests.

## Additional information

**Extended data** is available for this paper at https://doi.org/10.1038/s41588-025-02117-1.

**Correspondence and requests for materials** should be addressed to Richard Durbin.

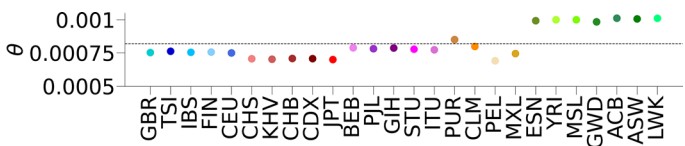

**Extended Data Fig. 1 | Heterozygosity estimates for the 26 individuals from distinct populations in the 1000 Genomes Project.** The scaled mutation rate $\theta = 4N\mu$ per population in the 1000 Genomes Project, using Waterson's estimator. The dashed, horizontal line indicates the mean. See Supplementary Table 1 for full information.

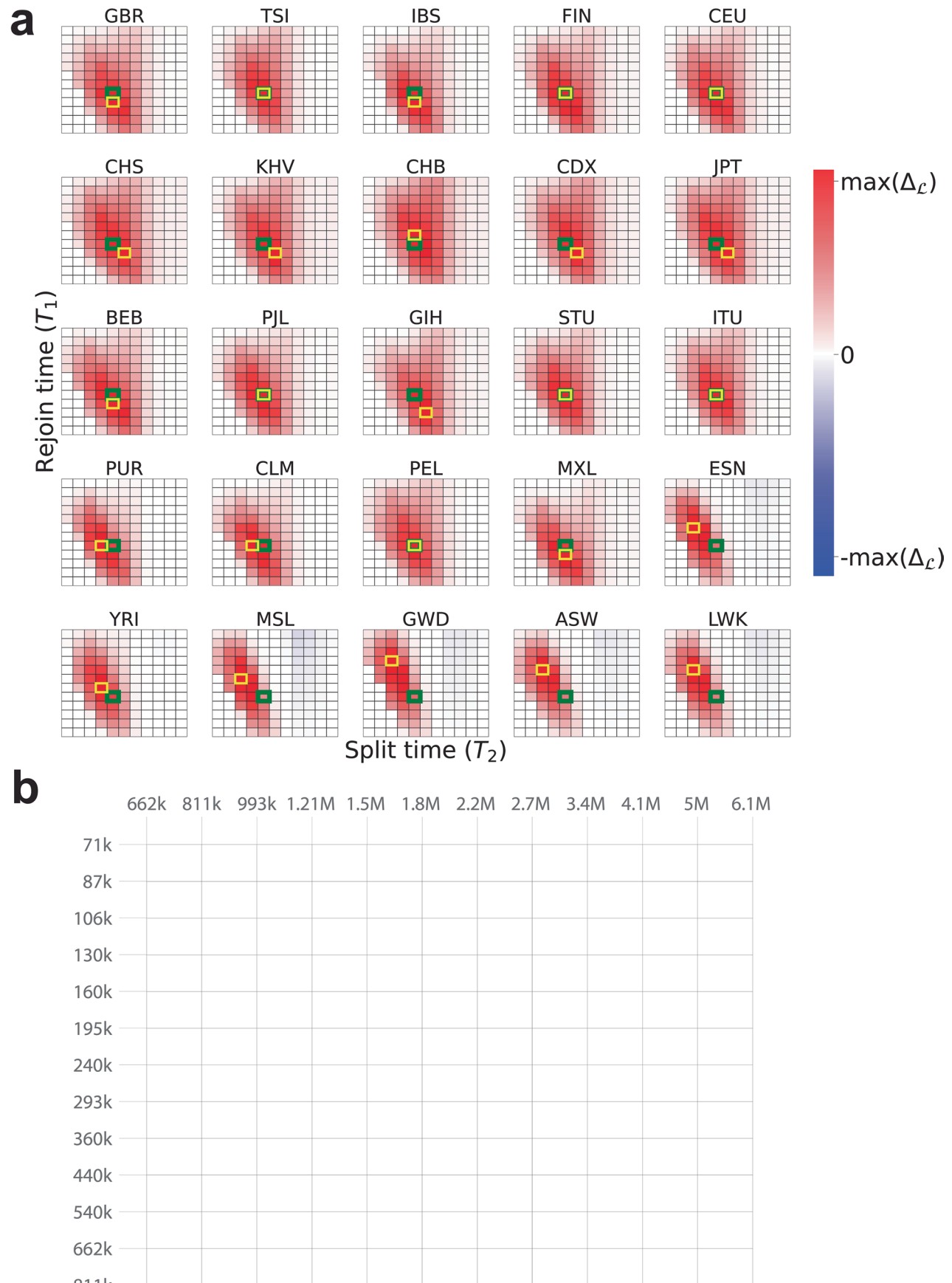

**Extended Data Fig. 2 | See next page for caption.**

**Extended Data Fig. 2 | Differences in log-likelihood between various structured models and the unstructured model.** (**a**) The log-likelihood difference ($\Delta_{\mathcal{L}} = \mathcal{L}_S - \mathcal{L}_U$) between cobraa and PSMC, for all given pairings of $T_1$ and $T_2$ for each population (ACB excluded). Each algorithm was run until convergence, defined as the change in $\mathcal{L}$ between subsequent iterations of the expectation-maximisation (EM) algorithm being less than one. Red indicates a positive difference, blue negative, and white zero. Each population is shown on its own scale. The green highlighted cell indicates the composite maximum likelihood (CML) estimate and the yellow highlighted cell indicates the maximum likelihood estimate per population. Higher on the y-axis indicates more recent $T_1$, and leftmost on the x-axis indicates more recent $T_2$. (**b**) A diagram indicating the corresponding time interval boundaries associated with each cell in **a**.

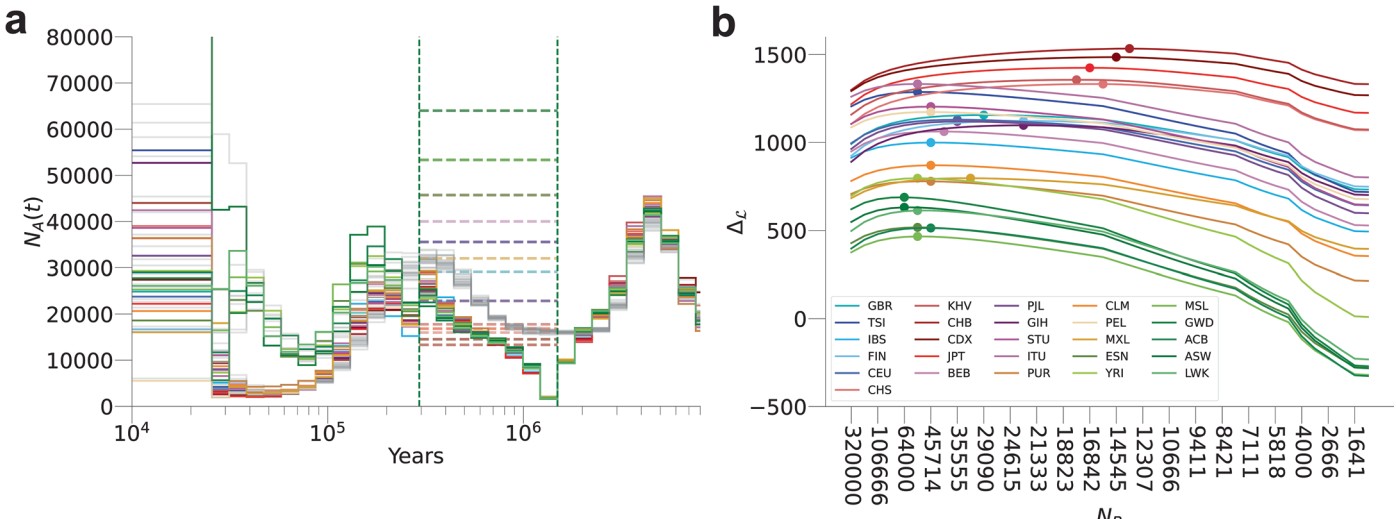

**Extended Data Fig. 3 | The inferred size of population B, for 26 individuals from the 1000 Genomes Project. (a)** Inference of $N_A(t)$ from cobraa on 26 populations from the 1000 Genomes project, with the inferred size of the ghost population $N_B(t)$ shown in the horizontal dashed lines. When optimizing we enforced that $N_B(t)$ must be constant due to identifiability problems. **(b)** Fixing the inferred $N_A(t)$, $\gamma$, $T_1$, $T_2$, and $\rho$ as shown in **a**, we compute the likelihood of the data with varying values of $N_B$, for each population. The plot shows the difference between this model ($\mathcal{L}_S(N_B)$) and the unstructured model ($\mathcal{L}_U$) in Fig. 4a, $\Delta_{\mathcal{L}} = \mathcal{L}_S(N_B) - \mathcal{L}_U$. The circles on each line represent the maximum likelihood estimate.

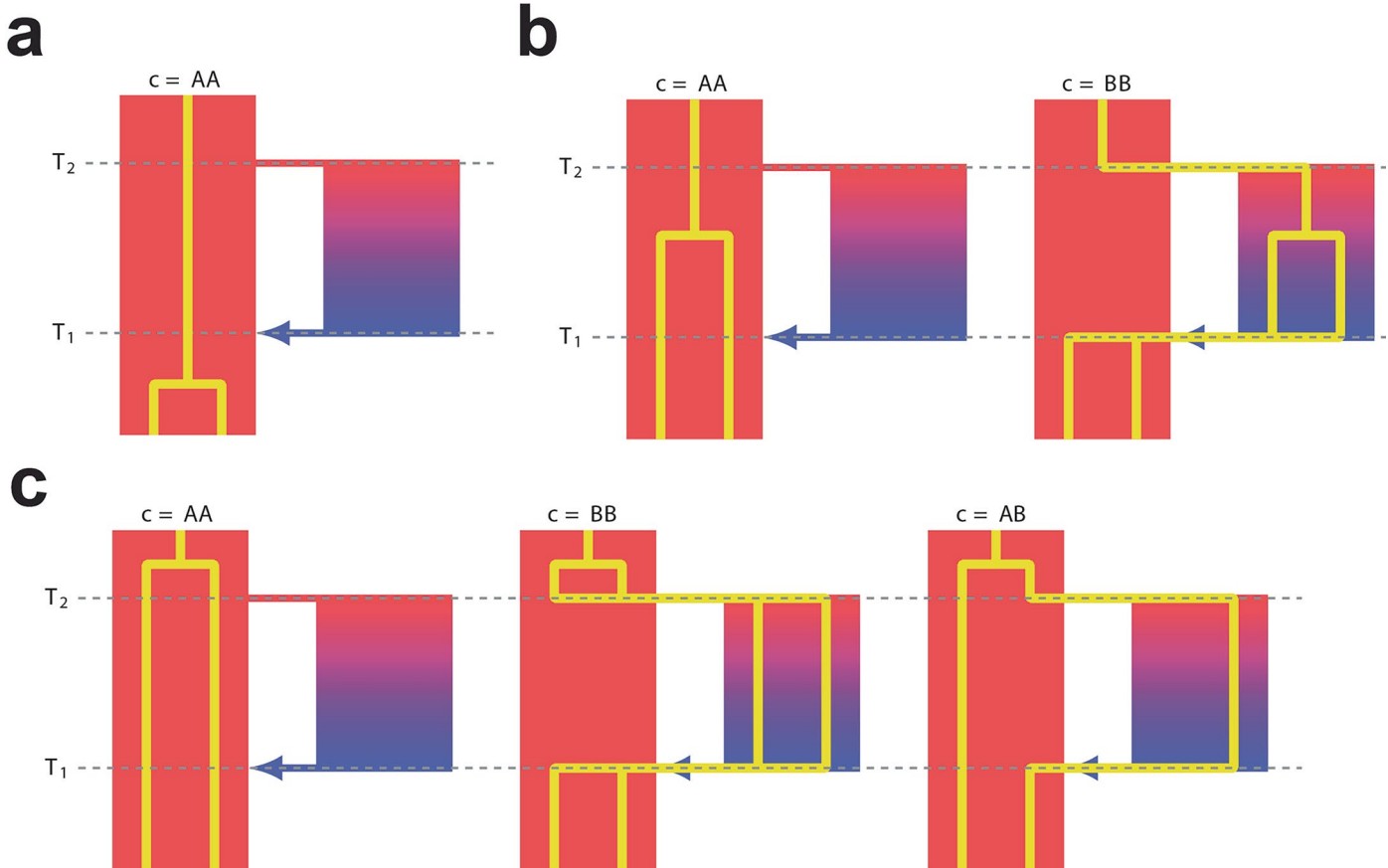

**Extended Data Fig. 4 | Diagram of the cobraa-path model.** A diagram of the possible ancestral lineage paths, $c$, as explicitly modeled in cobraa-path. (**a**) If the coalescence time was more recent than the admixture event ($T_1$), the only possibility is that the lineages coalesce in population A, which we denote as $c$ = AA. (**b**) If the coalescence time was in the structured period, between $T_1$ and $T_2$, then $c$ = AA or $c$ = BB. (**c**) If the coalescence time was more ancient than the split time, $T_2$, then $c$ = AA, $c$ = BB or $c$ = AB.

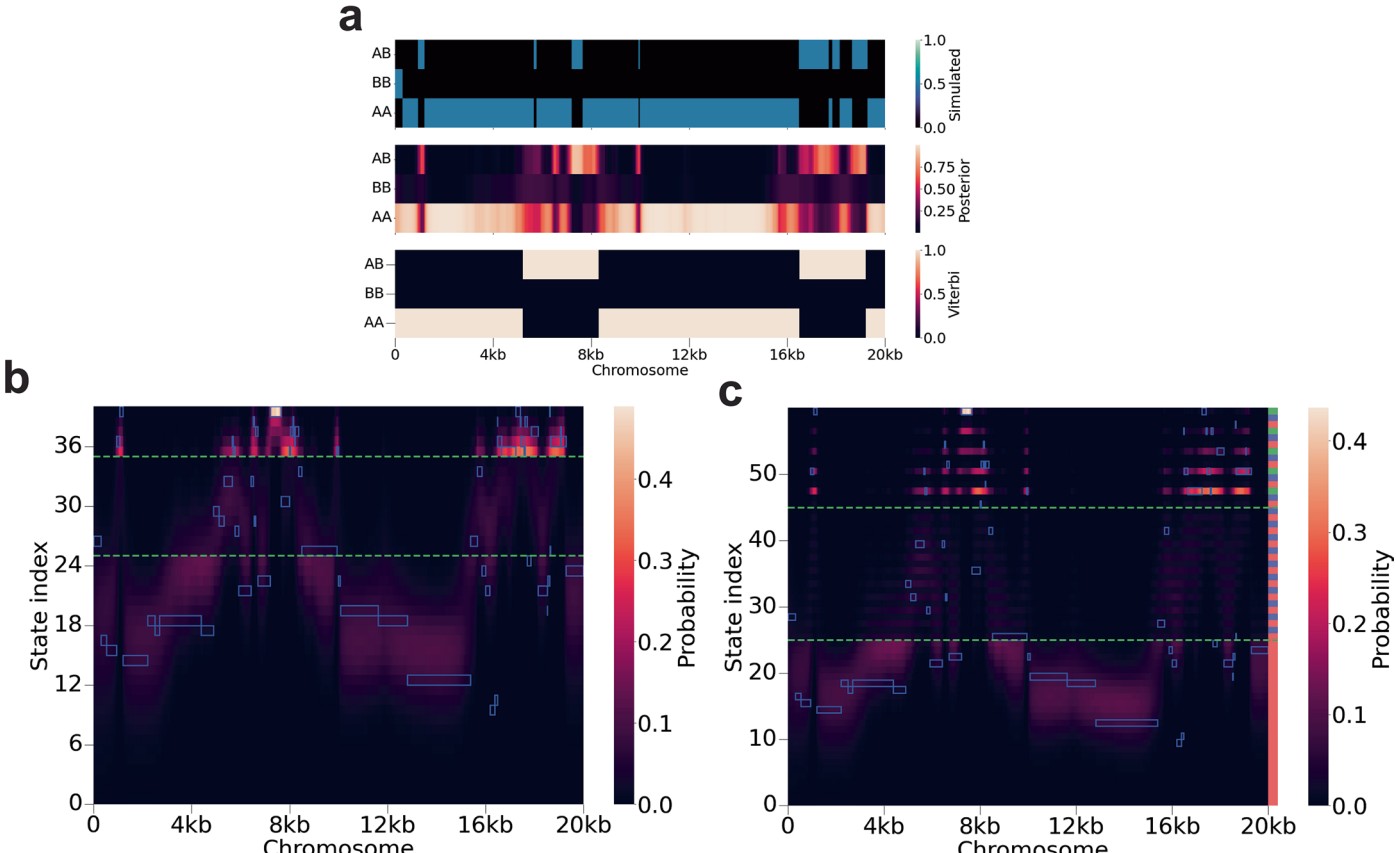

**Extended Data Fig. 5 | Posterior decoding from cobraa and cobraa-path, on simulated data.** We can decode the HMM of cobraa-path to infer the admixed regions of the genome (that is parts of the genome where the ancestral lineage pair went through AB or BB). The top panel of **a** shows the simulated lineage path across the genome, where the x-axis indicates the chromosomal position and the y-axis the ancestral lineage path. A structured model was simulated with $\mu/r = 1.25$, 40% admixture, and constant population sizes. Using the simulated structured parameters, the middle panel shows the marginal posterior probability of each lineage path (from the forward/backward algorithm), and the bottom shows the most likely lineage path (from the Viterbi algorithm). (**b**) The full cobraa decoding of the simulation (hidden states are discretized coalescence times), where the y-axis indicates the coalescence time with 0 being the present. The green, dashed, horizontal lines indicate the simulated split and admixture

times. (**c**) The full cobraa-path decoding of the simulation (hidden states are discretized coalescence times and ancestral lineage path). The y-axis indicates not only the coalescence time but also the ancestral lineage path, which is indicated by the shading in the right-most column. Red indicates AA, blue BB, and green AB. In the structured period, a red-blue pair indicates the same coalescence time, and more anciently than the split time a red-blue-green triple indicates the same coalescence time. The simulated states through the model are shown by the highlighted blue cells, and the posterior probabilities are indicated by the shading of the heatmap, with cream representing total confidence and black indicating no confidence. The y-axis in (a) indicates the ancestral lineage path, and in **b** and **c** the coalescence time, with more ancient states at the top and more recent ones at the bottom. The horizontal, green, dashed lines in **b** and **c** indicate the split and admixture time.

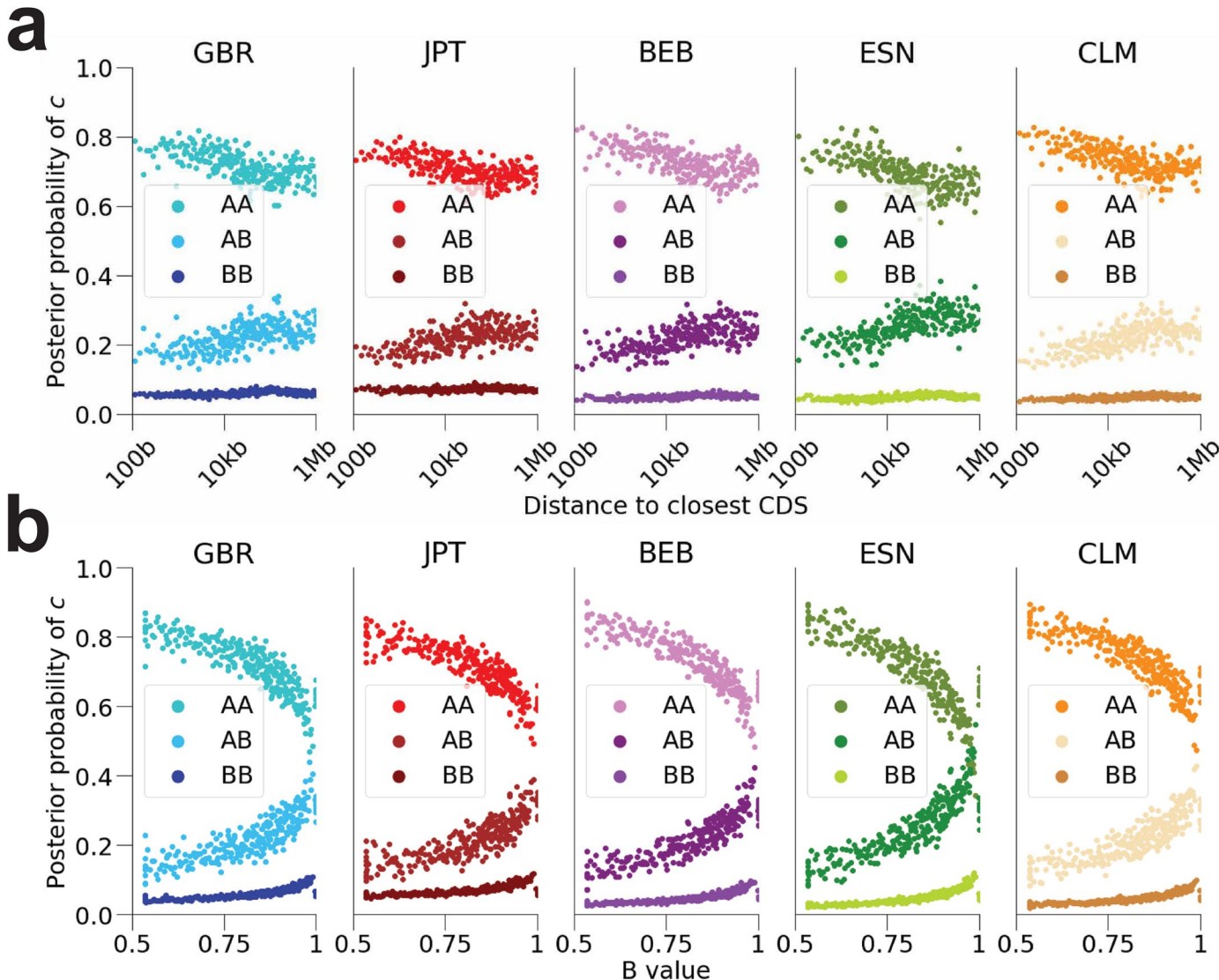

**Extended Data Fig. 6 | Correlation between the probability of an admixed lineage path and functional genomic information.** Correlation between the inferred probability of an admixed lineage path (conditional on the coalescence time being older than the admixture event) and the distance to coding sequence (CDS) (**a**) or strength of B-value (**b**). To get the inferred probability of an admixed lineage path, we decode the HMM of cobraa-path with the parameters as inferred from cobraa (Fig. 4b,c), and marginalize out the coalescence times. The Spearman correlation coefficients for each population are shown in Supplementary Table 2.

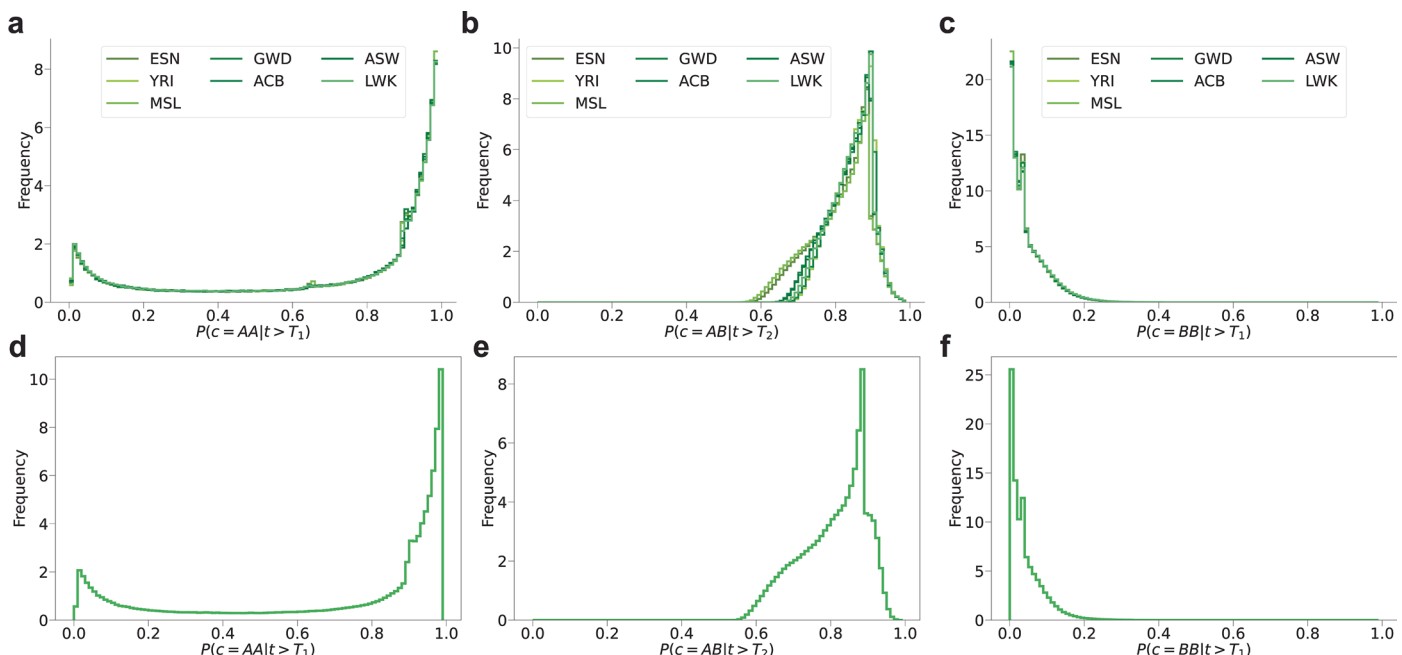

**Extended Data Fig. 7 | Density of posterior probability assignments to lineage combinations AA, AB, and BB, for real and simulated data.** (**a**–**c**) The density of posterior probability assignments to lineage combinations AA, AB, and BB for African samples analyzed using cobraa-path under our best fit structured model. (**d**–**f**) Corresponding plots for 1Gb of simulated data using parameters from the ESN model. (**b**,**e**) Given that $t > T_2$ there is a high confidence in assignment to AB, because most AA pairs coalesce prior to the $T_2$ due to the bottleneck at the founding of population A. (**c**,**f**) Although there are some real BB lineages (prior probability = 0.04) we do not have power to confidently identify these.

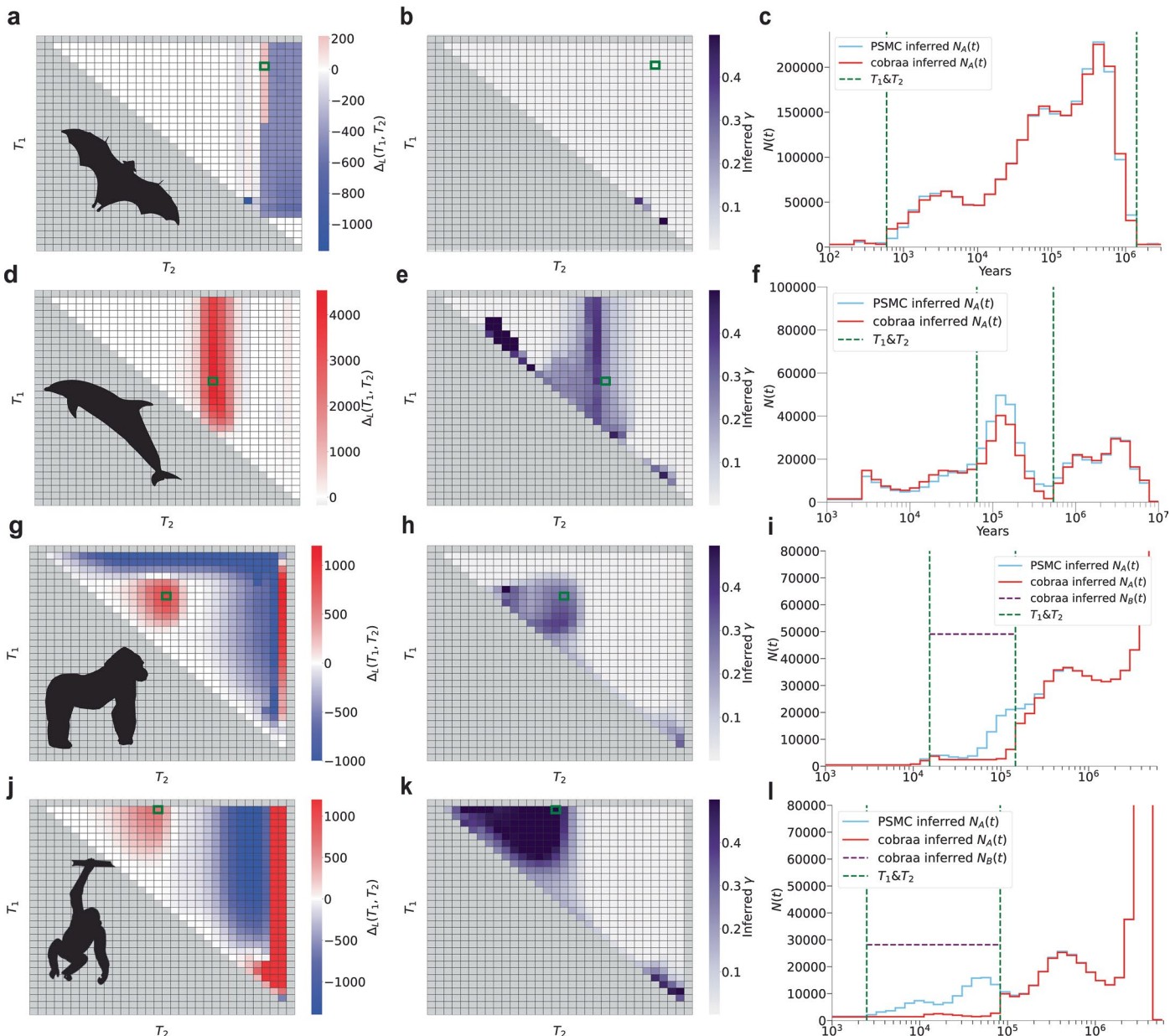

**Extended Data Fig. 8 | cobraa analysis on other species.** Inference from cobraa on various species. (**a**–**c**) Shown are parti-coloured bat, (**d**–**f**) common dolphin, (**g**–**i**) gorilla and (**j**–**l**) chimpanzee. (**a,d,g,j**) The leftmost column indicates the difference between the structured model for various time pairings and the unstructured model, $\Delta_{\mathcal{L}} = \mathcal{L}_S(T_1, T_2) - \mathcal{L}_U$. (**b,e,h,k**) The middle column indicates the inferred admixture fraction $\gamma$ from each time pairing. (**c,f,i,l**) The right column shows the inference of population size changes from PSMC and cobraa (the inferred size of $N_B$ in (**i**) and (**l**) was 690 ka and 164 ka, respectively).

# Reporting Summary

## Statistics

For all statistical analyses, confirm that the following items are present in the figure legend, table legend, main text, or Methods section.

| n/a | Confirmed | |
|---|---|---|
| ☐ | ☒ | The exact sample size (*n*) for each experimental group/condition, given as a discrete number and unit of measurement |
| ☐ | ☒ | A statement on whether measurements were taken from distinct samples or whether the same sample was measured repeatedly |
| ☐ | ☒ | The statistical test(s) used AND whether they are one- or two-sided *Only common tests should be described solely by name; describe more complex techniques in the Methods section.* |
| ☐ | ☒ | A description of all covariates tested |
| ☐ | ☒ | A description of any assumptions or corrections, such as tests of normality and adjustment for multiple comparisons |
| ☐ | ☒ | A full description of the statistical parameters including central tendency (e.g. means) or other basic estimates (e.g. regression coefficient) AND variation (e.g. standard deviation) or associated estimates of uncertainty (e.g. confidence intervals) |
| ☐ | ☒ | For null hypothesis testing, the test statistic (e.g. *F*, *t*, *r*) with confidence intervals, effect sizes, degrees of freedom and *P* value noted *Give P values as exact values whenever suitable.* |
| ☐ | ☒ | For Bayesian analysis, information on the choice of priors and Markov chain Monte Carlo settings |
| ☐ | ☒ | For hierarchical and complex designs, identification of the appropriate level for tests and full reporting of outcomes |
| ☐ | ☒ | Estimates of effect sizes (e.g. Cohen's *d*, Pearson's *r*), indicating how they were calculated |

*Our web collection on statistics for biologists contains articles on many of the points above.*

## Software and code

Policy information about availability of computer code

| Data collection | We used sequence data from the 1000 Genomes Project and HGDP. which is freely available online. |
|---|---|
| Data analysis | We used our new method. cobraa. the code for which is freely available: Rithub.com/trevorcousins/cobraa. |

For manuscripts utilizing custom algorithms or software that are central to the research but not yet described in published literature, software must be made available to editors and reviewers. We strongly encourage code deposition in a community repository (e.g. GitHub). See the Nature Portfolio guidelines for submitting code & software for further information.

## Data

Policy information about availability of data

All manuscripts must include a data availability statement. This statement should provide the following information, where applicable:
- Accession codes, unique identifiers, or web links for publicly available datasets
- A description of any restrictions on data availability
- For clinical datasets or third party data, please ensure that the statement adheres to our policy

We used publicly available sequence data from the 1000 Genomes Project and HGDP and provide links to access these in the Data and Code Availability section. We also now use publically available data from gorilla, chimpanzee, bat and dolphin, giving accession numbers.

# Research involving human participants, their data, or biological material

Policy information about studies with [human participants or human data](). See also policy information about [sex, gender (identity/presentation), and sexual orientation]() and [race, ethnicity and racism]().

| | |
|---|---|
| Reporting on sex and gender | We only use data from autosomes. |
| Reporting on race, ethnicity, or other socially relevant groupings | All human data we use comes from openly available reference genome resources (1000 Genomes Project, HGDP and the GRCh38 human reference), and we use the population identifiers provided by these resources. |
| Population characteristics | We refer to some populations as "African" and others as "non-African" |
| Recruitment | n/a |
| Ethics oversight | n/a - all data are previously published and fully open access |

Note that full information on the approval of the study protocol must also be provided in the manuscript.

# Field-specific reporting

Please select the one below that is the best fit for your research. If you are not sure, read the appropriate sections before making your selection.

☐ Life sciences    ☐ Behavioural & social sciences    ☒ Ecological, evolutionary & environmental sciences

For a reference copy of the document with all sections, see [nature.com/documents/nr-reporting-summary-flat.pdf](http://nature.com/documents/nr-reporting-summary-flat.pdf)

# Ecological, evolutionary & environmental sciences study design

All studies must disclose on these points even when the disclosure is negative.

| | |
|---|---|
| Study description | We fit a new model to various present day human sequences. We are studying the evolutionary past. |
| Research sample | Data from 1000 Genomes Project and Human Genome Diversity Project |
| Sampling strategy | Each of the 26 populations in the 1000 Genomes Project has numerous samples. We chose the first one of these listed per population. |
| Data collection | Data were downloaded from public resources in June 2022. |
| Timing and spatial scale | Present day sequences, from all over the world |
| Data exclusions | None |
| Reproducibility | Scripts for analysis and processing of data are given in github.com/trevorcousins/cobraa/tree/main/reproducibility |
| Randomization | No randomisation |
| Blinding | All processing was automated - no blinding was required. |

Did the study involve field work?    ☐ Yes    ☒ No

# Reporting for specific materials, systems and methods

We require information from authors about some types of materials, experimental systems and methods used in many studies. Here, indicate whether each material, system or method listed is relevant to your study. If you are not sure if a list item applies to your research, read the appropriate section before selecting a response.

## Materials & experimental systems

| n/a | Involved in the study |
|-----|----------------------|
| ☒ ☐ | Antibodies |
| ☒ ☐ | Eukaryotic cell lines |
| ☒ ☐ | Palaeontology and archaeology |
| ☒ ☐ | Animals and other organisms |
| ☒ ☐ | Clinical data |
| ☒ ☐ | Dual use research of concern |
| ☒ ☐ | Plants |

## Methods

| n/a | Involved in the study |
|-----|----------------------|
| ☒ ☐ | ChIP-seq |
| ☒ ☐ | Flow cytometry |
| ☒ ☐ | MRI-based neuroimaging |

## Plants

| | |
|---|---|
| Seed stocks | *Report on the source of all seed stocks or other plant material used. If applicable, state the seed stock centre and catalogue number. If plant specimens were collected from the field, describe the collection location, date and sampling procedures.* |
| Novel plant genotypes | *Describe the methods by which all novel plant genotypes were produced. This includes those generated by transgenic approaches, gene editing, chemical/radiation-based mutagenesis and hybridization. For transgenic lines, describe the transformation method, the number of independent lines analyzed and the generation upon which experiments were performed. For gene-edited lines, describe the editor used, the endogenous sequence targeted for editing, the targeting guide RNA sequence (if applicable) and how the editor was applied.* |
| Authentication | *Describe any authentication procedures for each seed stock used or novel genotype generated. Describe any experiments used to assess the effect of a mutation and, where applicable, how potential secondary effects (e.g. second site T-DNA insertions, mosiacism, off-target gene editing) were examined.* |

