## [Peer Review File · Nature Genetics]

A structured coalescent model reveals deep ancestral structure shared by all modern humans

Corresponding Author: Dr Richard Durbin

Version 0:

Decision Letter:

24th May 2024

Dear Dr Durbin,

Your Article, "A structured coalescent model reveals deep ancestral structure shared by all modern humans" has now been seen by 2 referees. You will see from their comments below that while they find your work of interest, some important points are raised. We are interested in the possibility of publishing your study in Nature Genetics, but would like to consider your response to these concerns in the form of a revised manuscript before we make a final decision on publication.

In brief, the two reviewers are positive for the aims of your work, and will likely support publication once clarifications have been provided for their specific comments.

Referee #1, while saying cobraa is "very interesting", has one very important concern on an ancient and repeatedly-observed 'hump' which their report suggests may be a result of the coalescence model, rather than reflecting true biology. They ask for some specific further analyses to clarify this point, and make similar constructive critiques of a few other aspects of your work.

Reviewer #2 is straightforwardly positive; their requests are primarily minor but are important as they would clarify the technical details of cobraa. Some additional analysis is also requested.

In our reading of these reports, we think the various specific comments and the requested revisions are important and, to our understanding, not impossible. We would, unsurprisingly, highlight Referee #1's major concern as especially important to address: it seems that further exploration of the cause of these 'ancient humps' may well lead to results that will be of interest to anyone applying this kind of coalescence model, thus broadening the appeal and impact of your study.

To guide the scope of the revisions, the editors discuss the referee reports in detail within the team, including with the chief editor, with a view to identifying key priorities that should be addressed in revision and sometimes overruling referee requests that are deemed beyond the scope of the current study. We hope that you will find the prioritized set of referee points to be useful when revising your study. Please do not hesitate to get in touch if you would like to discuss these issues further.

We therefore invite you to revise your manuscript taking into account all reviewer and editor comments. Please highlight all changes in the manuscript text file. At this stage we will need you to upload a copy of the manuscript in MS Word .docx or similar editable format.

*2) If you have not done so already please begin to revise your manuscript so that it conforms to our Article format instructions, available

[here](http://www.nature.com/ng/authors/article_types/index.html).

*3) Include a revised version of any required Reporting Summary: <https://www.nature.com/documents/nr-reporting-summary.pdf>

Link Redacted

Sincerely,

Michael Fletcher, PhD
Senior Editor, Nature Genetics
ORCID: 0000-0003-1589-7087

Referee expertise:

Referee #1: genetics, population genetics, including methods development.

Reviewers' Comments:

Reviewer #1:

Remarks to the Author:

This is a very interesting manuscript reporting the development of a tool for identifying ancient population admixture. Using this tool, the authors infer that modern humans stem from an 80/20 admixture of two populations that diverged more than a million years ago. This finding is reminiscent of a hypothesis that dates back to the original publication of the PSMC, in which Li and Durbin hypothesized that a population size "hump" occurring during early human history was actually a period of ancient population structure. The details are a bit different since the period of structure inferred in this new paper is older than the one hypothesized in the 2011 paper, and inclusion of this structure does not seem to make the ancient population size hump go away.

Unfortunately, a major concern I have is that the ancient history inferred by PSMC seems in general less reliable than the recent demography that is inferred by this method, as explored for instance by Beichman, et al. G3 2017. It's no specific shortcoming of PSMC that ancient history is harder to infer than recent history is, but the Beichman, et al. paper found that PSMC's inferences of population size changes in the ancient past did not seem as reliable as other programs' more regularized inferences in these time regimes. There is also ample anecdotal evidence, thanks to the widespread application of PSMC to non-model species, that all PSMC histories have a characteristic shape, particularly in the ancient past, with a humped shape reminiscent of the shape that led to the ancient structure hypothesis in the 2011 paper. This makes me question the idea that the paper's finding reveal something distinctive about human history rather than something more universal about how real population genomic data differs from the output of unstructured coalescent simulations.

To test my recollection that ancient humps are a near-universal feature of PSMC histories, I did a google scholar search for PSMC and found this sort of shape in each of the first 5 non-hominoid data analysis papers that came up in the search (collared flycatchers, Nadachowska-Brsyska et al 2016; grouse, Kozma et al 2018; apples, Sun, et al. 2020; pigs, Groenen et al. 2012; cannabis, Ren et al 2021). It might be the case that ancient population structure was present in the histories of all of these other species—after all, historical admixture has been detected in lots of different populations since the publication

of the Neanderthal genome made gene flow a hot topic. However, it's also possible that PSMC and cobraa are simply very sensitive to the fact that all real populations have some amount of substructure. Given that ideal Wright-Fisher populations do not exist, maybe a structured model will essentially always fit the data better than a nested unstructured model, especially given that the authors do not seem to use anything like the AIC or BIC to penalize the unstructured model for having fewer parameters.

I would not normally think it was reasonable to suggest that a paper needs to be expanded to include analyses of other species, but in this case I think an analysis of at least one or two other species is needed to inform interpretation of the human data. If cobraa tends to find that ancient population structure is a better fit than panmixia in nonhuman species as well, I think that finding would be important and impactful to publish but would be more of a story about methodology rather than the story about human history being currently presented here. Theoretical population geneticists are always asking whether the ancient humps in PSMC plots should be interpreted literally, and empirical users of demographic inference methods are always uncritically interpreting these trajectories as reflections of events like the last glacial maximum; more general insight into what shapes ancient PSMC trajectories would help resolve these conflicting narratives. If instead this ancient structure appears unique to humans, the absence of similar inferences about other species would give me more confidence that populations A and B really existed as discrete entities and reflect something distinctive about human history in particular.

It would also be informative to run cobraa on one or more of the high coverage archaic hominid genomes because of this paper's implicit hypothesis that we should not see similar population structure in their histories (or at least that the contribution of population B should be lower than is seen in humans). One thing I noticed about the PSMC plot in the Prüfer, et al. Altai Neanderthal paper is that the Altai Neanderthal population size trajectory appears to line up with the human population size trajectory during the time period when A and B appear to diverge, which is not what we would expect if Neanderthals were entirely an offshoot of population A.

The aspect of this paper that I find most compelling is the fact that the authors are able to classify specific pieces of genomic material as candidate tracts of introgression from "population B." This posterior decoding opens up many more avenues for sanity checking the authors' hypothesis than is possible in other studies that have used features of one- or two-locus site frequency spectra to argue for the existence of ancient population structure in Africa. This decoding allows the authors to test regions of A and B ancestry for divergence to Neanderthals and Denisovans, and they find that Neanderthals and Denisovans appear more related to segments of B ancestry. However, one thing that gave me pause about this analysis is the numerical range of the probabilities $P(c = BB | t = T_1)$: it looks like there are very few regions where $P(c = BB | t = T_1)$ is higher than 25%, and archaic/human divergence seems to level off starting at $P(c = BB | t = T_1) = 0.05$. If there are 5 times more B/B tracts in the set of regions with 25% posterior probability compared to the set of regions with 5% posterior probability, shouldn't the set of regions with 25% posterior probability look significantly more diverged from Neanderthals compared to the regions with 5% posterior probability? In contrast, $P(c = AA | t > T_1)$ seems to span the full range from 0 to 1, with lots of regions where $P(c = AA | t > T_1)$ is near 0%. Can the authors offer some insight into why these distributions are so different? I am worried that if the model is so biased toward classifying regions as "AA" rather than "BB", there might be many systematic differences between AA and BB regions that affect downstream analyses. For example, maybe regions with recent coalescence times close to T_1 are more likely to be confidently classified as AA and are also more likely to be near genes compared to regions with much older coalescence times. When you simulate data from a history like the one inferred here, do you get such different distributions of AA and BB posterior probabilities?

Figure S14 shows some differences among human populations and archaic genome divergence that seem potentially important but don't appear to be to square with the narrative about Neanderthals and Denisovans branching off from population A. This population variation should be highlighted in the main text and discussed in relation to the model the paper is putting forward. The first feature of S14 that jumped out at me is the qualitative difference between $P(c = BB | T_1 < t < T_2)$ and $P(c = BB | t > T_2)$ —assuming that the red lines are Asian populations, it looks like regions with very old coalescence times and a very low probability of BB ancestry are deeply diverged from Neanderthal/Denisova in East Asians, close to Neanderthal/Denisova in Africans and Amerindians, and intermediately diverged in Europeans? Then there's an opposite and less dramatic ancestry gradient when you look at regions with relatively high probability of BB ancestry, where divergent to Neanderthal/Denisova is lowest for East Asians and highest for Africans. In contrast, variation among populations is low when you look at the relationship between archaic divergence and the posterior probability of AA ancestry. This seems like the opposite of what you'd expect if Neanderthals and Denisovans branched off from population A: in Europeans and East Asians, AA ancestry should on average have more archaic affinity than AA ancestry in Africans, since some of the "A" ancestry in non-African populations should be tracts of archaic introgression. In contrast, "B" ancestry in all populations should be equally diverged from Neanderthals and Denisovans. Maybe my interpretations are off base, but I think these nonintuitive population differences need to be highlighted in the main text and given some explanation.

Minor comments:

Lines 36-37 say, "for any panmictic model...there necessarily exists a structured model that can generate exactly the same coalescence rate profile." Isn't this trivially true? Did you mean to say the converse, that for every structured model there's a panmictic variable size model with the same rate profile? Or is the idea that you can recapitulate the rate with structure without having to introduce population size changes?

The legend of Figure 2 is hard to parse because alpha and beta aren't defined. I think they're supposed to be the population size change times but if so that should be annotated in panel a.

At the start of section 3.2, the specification of the structured hypothesis isn't very well explained. Is the idea to specify a history containing one period of structure with flexible start and end times and allow the model to find T_1 and T_2 in a completely unsupervised way, or is there something encoded in the model that encourages T_1 and T_2 to occur near the start and end points of the ancient population size hump?

Reviewer #2:

Remarks to the Author:

Review of "A structured coalescent model reveals deep ancestral structure shared by all modern humans" by Trevor Cousins, Aylwyn Scally, and Richard Durbin

The author introduce a novel method that can detect structure in the populations ancestral to contemporary individuals. Specifically, the method is an extension of the widely use method PSMC that infers past changes in effective population size from a single diploid genome of a contemporary individual. It has been noted in the literature that different models of structure in the ancestral population can lead to the same profile of inferred ancestral population sizes, when assuming a single panmictic ancestral population. The authors thus extend the framework of PSMC to incorporate structure in the ancestral population, in particular, a model of a single population that splits into two, which subsequently admix at given fractions to form the main lineage of modern human populations.

The authors apply their method to individuals from different populations from the 1000G dataset and the HGDP dataset, and infer that the most likely model is a deep split of an ancestral population ~1.5 million years ago, and the two ancestral populations subsequently admix around 300k years ago, with a ratio of 80%:20%. In addition, the authors augment the hidden state of their model to identify the ancestral population that a particular genomic region in a contemporary individual derives from. Upon investigating region-specific divergence of modern humans and Neanderthals, the authors provide evidence that the major ancestral population (80%) was likely the ancestor of Neanderthals. The authors also perform a GO analysis on the identified regions to investigate their functional importance. In addition, the authors provide evidence that the minor ancestry was selected against.

The manuscript is a timely contribution to efforts aimed at understanding structure in populations ancestral to modern humans, particularly in the context of ghost-populations that left no contemporary descendants, and how this is related to known archaic populations like Neanderthals and Denisovans. The manuscript is well written, the premise clearly explained, the description of the method easy to follow, and the thoughtful analysis provides clear evidence to support the conclusions. I do not have any major points of criticism. However, I have a couple of comments that I think should be addressed to further strengthen the manuscript, and make it accessible to a wider audience.

In particular:

- When reading only the main text, some notation is not well introduced. Some examples include:

- p.4, l.77: The authors write "we now define...", but the definition is in the supplement. Please introduce Q^S and Q^U conceptually here.

- p. 7, l.140: L_S and L_U are not defined. Please define.

- p.3, l.61: Add: "In Section 9.3.3 of the Supplement, we show that identifiability issues arise when both the sizes of both ancestral population can change, which justifies our choice to set N_B constant." <-- I think it would be good to mention this here already.

- p.5, l.106: The ratio of μ/r is given, but neither a value of μ or r . Please specify. Similarly in the caption of Figure 3. And p.5, l.116.

- Figure 3: In (a), the simulated T_1 and T_2 match the values later inferred for human data approximately. However, for (b), T_1 and T_2 are substantially lower. Was this choice to better exhibit the differences visually? Please provide a simulation like 3(b) where T_1 and T_2 match the values inferred from real data (approximately). In addition, also provide a simulation study where you simulate data using the $N_A(t)$ (and other parameter) inferred from the real data (and matching μ and r), and present the estimates. This would indicate how robust the inference from the real data is. If there is a reason either of those cannot be provided, please state the reason.

- Figure 3(d): Please indicate ticks and the range of values on the x- and y-axes. Does the white-ish square indicate the true value? If so, please indicate more clearly, perhaps a stronger color contrast and bolder outline. Also, is there a way to better the scale of the colorbar then putting $-1.6e6$ at the top?

- I was somewhat surprised that in the model selection simulations displayed in Figure 3(e), the difference in log-likelihood for the unstructured simulations can be negative. The author claim (and I think it is true) that the structured model is nested in the unstructured model. In that case, shouldn't the likelihood difference between the true maxima always be non-negative? The method certainly requires discretization, numerical optimization, and the EM algorithm is only guaranteed to find local optima. Are these the only factors that make the difference negative? Is there some subtlety that I am missing here? Is a difference of -100 cause for concern that the numerical optimization might not identify a "good" optimum? Could you please add a short discussion of these issues?

- Somewhat related: Figure 4(c) shows that the log-likelihood difference for the real data is 500 or more. The structured model has more parameters than the unstructured model, so some positive difference is expected. I do not believe that the expected difference will be anywhere near 500 (especially since N_B is assumed constant), and thus, the results hold. But please add a short discussion of the difference in number of parameters, and why it is no cause for concern.

- Figure 4(b): The authors indicate the panmictic population size inferred by PSMC in grey. When looking at this plot, I was also very curious about the inferred N_B . N_B is indicated in Suppl. Figure 6, but I was wondering if there a good way to integrate it in Figure 4(b), or maybe even provide it in a separate panel in Figure 4.

- p.7, l.154: The authors indicate that they use PSMC to obtain estimates of $N_A(t)$ in the panmictic case. I suspect that they do not literally run the software PSMC, but rather cobraa, with the parameters restricted to resemble PSMC. Could you please indicate explicitly whether you run the software PSMC, or cobraa restricted to mirror PSMC.

- For some of the functional analysis of the admixed regions, the p-values are very strong. This makes me wonder whether chromosomal linkage is correctly accounted for. Specifically, on p.9, l.211, correlation between dcCDS and probability of admixture is computed. However, the probabilities of admixture are auto-correlated along the genome. Is this accounted for when computing the significance of the correlation? If so, how? Furthermore, on p.9, l.225, the significance of a set of genes to overlap extreme values of admixture is computed. The statistic for extremity $H(x)$ is computed in increments of 1kbp. However, genes can be longer than 1kbp, and thus each gene can "try" multiple times to overlap with an extreme position. I wonder whether a binomial with $p=0.01$ is thus the right null model. Also, $H(x)$ is auto-correlated. I do believe that the signals the authors report are significant, but perhaps the p-values could be inflated? Please clarify how these issues are accounted for, if they are indeed issues.

- p.14, l.366: Please indicate that the a detailed derivation can be found in the supplement.

- p.15, l.412: Did you use population-specific B-maps?

- p.17, l.465: Please indicate more clearly that the only difference between the analysis in this section and the analysis in the main text is that the each chromosome is cut into two "independent" pieces. The nucleotides marked as missing are exactly the same as before. This is a bit hard to follow.

- p.18, l.516: How is the divergence calculated? Between only the focal individual and Neanderthal? Or all individuals in the focal population?

- Supplementary Figure 4: As for Figure 3(d), please indicate values on the axes and indicate the truth more clearly.

- Supplementary Figure 15: For some populations, the MLE of T_1 and T_2 is on the edge of the grid used, indicating that the true MLE could lie outside of the grid. This effect would lead to underestimating the true maximum of the CML. I think it would be appropriate to extend the grid such that all individual maxima are truly in the interior. Please do so or comment why this is not necessary.

- The notation in the derivation of the transitions in Section 9.3 is somewhat inconsistent, which makes it somewhat hard to follow. I think it would be good to streamline this notation for interested readers:

- All derivations assume $t \in \{\tau_\alpha, \tau_{\alpha+1}\}$ and $s \in \{\tau_\beta, \tau_{\beta+1}\}$. Please state this clearly, perhaps on p.53, l.946, otherwise it is hard to follow why α appears in section 9.3.1 and subsections.

- T_α is never defined to be equal to τ_α . Thus, all instances of T_α and T_β in section 9.3.1 and subsections have to be replaced by τ_α and τ_β . Or the notation made more explicit. The notation T_α is also somewhat ambiguous if T_1 and T_2 exist.

- The notation λ_{A_α} should be properly introduced.

- Many ψ have a subscript β in addition to the case number. This seems obsolete.

- Supplementary Figure 27: These figures are a bit hard to interpret: No scale is indicated for the likelihood. Please indicate the scale. Furthermore, the disconnected regions in the likelihood-surface can be an artifact of the discretization grid for the values on the x and y axis. While the likelihood surfaces are shallower in the middle, the difference from one contour line to the next could still be 100 log-likelihood units, which I don't think would be fair to be interpreted as "unidentifiable."

Minor points:

- p.3, l.67, "... the coalescence times of the two haploid sequences across the genome."

- p.4, l.81: "Moreover, the relative difference increases when the admixture ... between A and B increases."

- Figure 3(e): Please place a legend in the Figure that indicates the color coding.

- p.5, l.137: Please explicitly indicate that all parameters in this section, except the split times, are as in the previous paragraphs (What is μ/r ?).

- p.8, l.154: "... unstructured (PSMC) (4a) and structured (cobraa) (4b) model ..."

- p.9, l.209: Define CDS.

- p.9, l.220: What does "inferred admixture" mean? Does it mean inferred B ancestry?

- Figure 5: The period at the end of subcaption (a) is weirdly spaced and bold. In subcaption (b), "increases" has a bold i.

- Supplementary Figure 1, Caption: "Relative difference in transition matrices for different model parameters."

- Supplementary Figure 1, Caption: "For the matrix ξ , we plot values in the rows that corresponds to ..."
- Supplementary Figure 2, Caption: "(a) Data are the sequence of true simulated coalescent times ..."
- Supplementary Figure 2, Caption: Third to last line, the S is not a subscript for L_S.
- Supplementary Figure 14: Please indicate the values on the y-axes.
- p.49, l.835: Subscript missing for x: " $x_i = 0$ ", " $x_i = 1$ ", and " $x_i = -1$ ".
- p.50, l.869: Doesn't Cardin and McVean's model allow 1 and 3, but not 2?
- p.51, l.891: "... in MSMC2 [99] Malaspinas et al. integrate ..."
- p.51, l.893: "Similarly, to create the transition matrix ..., we follow Malaspinas et al. and take ..."
- p.52, l.917: disallowed
- p.54, Equation (21): C^{AB} in numerator.
- p.55, Equation (22): In the last line, I think the γ in front of the parentheses is wrong.
- p.55, Equation (23): Replace T_{j+1} with τ_{j+1} (Same in (24) and (25))
- p.55, l.965: " $S \in \{1, \dots, D-1\}$ "
- p.55, Equation (26): Last line: T_S is incorrect. Replace λ_{A_b} by λ_{A_α} . Replace t_2 by t . Replace T_b by T_α .
- p.56, Equation (27): Replace T_S with T_1 .
- p.56, l.971: "Similar to Malaspinas et al. in [99] and Schiffels and Durbin in [38], we use the expected coalescent time $\langle \dots \rangle$ in interval β (see ...)." \leftarrow No integration is performed here.
- p.56, Equation (30) and (31): I believe there has to be $1/\lambda$ before the inner parentheses (in addition to the one before the outer parentheses).
- p.56, l.984: The notion of a "solid lineage" is never introduced.
- p. 57, Equation (36) and (37): In all denominators, t has to be replaced by s . These ψ are also functions of s , so please indicate that.
- p.58, Equation (39): Replace T_S with T_1 .
- p.58, Equation (40): Some of the "L" have a dash-through.
- p.57, Equation (48) and (49): In all denominators, λ has to be replaced by $\lambda(s)$. These ψ are also functions of s , so please indicate that.
- p.67, l.1116: The number of parameters has to use the index of T_1 and T_2 , not the times directly.

Version 1:

Decision Letter:

23rd Sep 2024

Dear Richard,

Your Article, "A structured coalescent model reveals deep ancestral structure shared by all modern humans" has now been seen by 2 referees. You will see from their comments below that while they find your work has improved in revision, there are still a few important points raised. We continue to be interested in the possibility of publishing your study in Nature Genetics, but would like to consider your response to these concerns in the form of a revised manuscript before we make a final decision on publication.

Briefly, Referee #1 is now satisfied and has no further requests. Reviewer #2 has some minor points - some more substantive than others - and we would like to make an editorial check on your responses to these comments before deciding whether further review is indeed required, as their report suggests may not be.

To guide the scope of the revisions, the editors discuss the referee reports in detail within the team, including with the chief editor, with a view to identifying key priorities that should be addressed in revision and sometimes overruling referee requests that are deemed beyond the scope of the current study. We hope that you will find the prioritized set of referee points to be useful when revising your study. Please do not hesitate to get in touch if you would like to discuss these issues further.

We therefore invite you to revise your manuscript taking into account all reviewer and editor comments. Please highlight all changes in the manuscript text file. At this stage we will need you to upload a copy of the manuscript in MS Word .docx or similar editable format.

*2) If you have not done so already please begin to revise your manuscript so that it conforms to our Article format instructions, available

[here](http://www.nature.com/ng/authors/article_types/index.html).
Refer also to any guidelines provided in this letter.

*3) Include a revised version of any required Reporting Summary: <https://www.nature.com/documents/nr-reporting-summary.pdf>

Link Redacted

We hope to receive your revised manuscript within four to eight weeks. If you cannot send it within this time, please let us know.

Sincerely,

Michael Fletcher, PhD
Senior Editor, Nature Genetics
ORCID: 0000-0003-1589-7087

Reviewers' Comments:

Reviewer #1 (Remarks to the Author):

The authors have done a commendable job with their thoughtful and extensive revisions. I was surprised that this expert team ran into problems trying to reproduce prior PSMC results on archaic genomes, but the paper's other new analyses, particularly the new analyses of non-human genomes, have largely alleviated my skepticism about the authenticity and interpretability of the ancient structure that is being found. The nonhuman results are fascinating in their own right and seem likely to inspire useful new studies of the ubiquity of admixture in nature.

Reviewer #2 (Remarks to the Author):

Second review of "A structured coalescent model reveals deep ancestral structure shared by all modern humans" by Trevor Cousins, Aylwyn Scally, and Richard Durbin

The authors have addressed most of my concerns satisfactorily. However, I do think that some minor points remain to be addressed, which I will list below (Page and lines refer to new revised paper). Addressing these points does not affect any of the major results and should be fairly quick. I thus do not think that these necessitate another round of review and could perhaps just be checked by the editor once implemented (and given that there is no major disagreement).

- p.16, l.455: I believe that the enrichment analysis for AAG and ASG vs H(x) is still not entirely correct. The authors claim correctly that 10,632 out of 21,587 1%-H(x) positions (48%) should overlap with genes. But the authors then incorrectly use 680 AAGs or 1287 ASGs to compute the p-value of the enrichment. The value that should be used instead of the number of genes is the number of positions that overlap with these genes, which, judging from Supplementary Figure 8 are likely more than 1 for each gene. Thus the 15-fold enrichment and the p-value of $< 1e-100$ is inflated.

- Figure 3: The labels for the panels are out of order.

- p.10, l.243: In the main text, the authors state that PANTHER is used for the GO analysis, whereas in the Methods (p.16, l.466) the authors indicate that geneontology.org is used. Please clarify which program is used for the GO analysis.
- Section 9 "Supplementary Text" on p.36 is empty.
- p.60, l.1095: Remove the duplicate "and".
- p.60, l.1096: s is not integrated out, it is set to the mean value in the respective interval. Please correct.
- p.61, Equation (35), (36), p.64, Equation (47), (48): Replace s on the right-hand side by $\langle t_{\beta} \rangle$.
- p.16, l.449: Please indicate in the manuscript that the YRI B-maps were used, and that this is reasonable since the correlation between different population B-maps is very high.
- Supplementary Figure 4: Please add ticks on the x- and y-axis to the figure.
- p. 78, l. 1335: I do not agree with the authors that Supplementary Figure 24 provides evidence for multiple local maxima. Yes, a lot of cells are yellow-ish, but that makes it just impossible to deduce from the plots how many maxima there are. I think that statement should be rephrased a bit more carefully.

Version 2:

Decision Letter:

Our ref: NG-A65040R1

22nd Oct 2024

Dear Richard,

Thank you for submitting your revised manuscript "A structured coalescent model reveals deep ancestral structure shared by all modern humans" (NG-A65040R1).

We've made an editorial check of the changes made from the last version and we are satisfied that these do not need further peer review. Therefore, we'll be happy in principle to publish your work in Nature Genetics, pending minor revisions to satisfy any final requests and to comply with our editorial and formatting guidelines.

Thank you again for your interest in Nature Genetics. Please do not hesitate to contact me if you have any questions.

Sincerely,

Michael Fletcher, PhD
Senior Editor, Nature Genetics
ORCID: 0000-0003-1589-7087

Cobraa paper: Response to reviewers

Below we include the reviewers' comments in black and our responses in blue.

Reviewer #1:

Remarks to the Author:

This is a very interesting manuscript reporting the development of a tool for identifying ancient population admixture. Using this tool, the authors infer that modern humans stem from an 80/20 admixture of two populations that diverged more than a million years ago. This finding is reminiscent of a hypothesis that dates back to the original publication of the PSMC, in which Li and Durbin hypothesized that a population size “hump” occurring during early human history was actually a period of ancient population structure. The details are a bit different since the period of structure inferred in this new paper is older than the one hypothesized in the 2011 paper, and inclusion of this structure does not seem to make the ancient population size hump go away.

Unfortunately, a major concern I have is that the ancient history inferred by PSMC seems in general less reliable than the recent demography that is inferred by this method, as explored for instance by Beichman, et al. G3 2017. It's no specific shortcoming of PSMC that ancient history is harder to infer than recent history is, but the Beichman, et al. paper found that PSMC's inferences of population size changes in the ancient past did not seem as reliable as other programs' more regularized inferences in these time regimes. There is also ample anecdotal evidence, thanks to the widespread application of PSMC to non-model species, that all PSMC histories have a characteristic shape, particularly in the ancient past, with a humped shape reminiscent of the shape that led to the ancient structure hypothesis in the 2011 paper. This makes me question the idea that the paper's finding reveal something distinctive about human history rather than something more universal about how real population genomic data differs from the output of unstructured coalescent simulations.

To test my recollection that ancient humps are a near-universal feature of PSMC histories, I did a google scholar search for PSMC and found this sort of shape in each of the first 5 non-hominoid data analysis papers that came up in the search (collared flycatchers, Nadachowska-Brsyska et al 2016; grouse, Kozma et al 2018; apples, Sun, et al. 2020; pigs, Groenen et al. 2012; cannabis, Ren et al 2021). It might be the case that ancient population structure was present in the histories of all of these other species—after all, historical admixture has been detected in lots of different populations since the publication of the Neanderthal genome made gene flow a hot topic. However, it's also possible that PSMC and cobraa are simply very sensitive to the fact that all real populations have some amount of substructure. Given that ideal Wright-Fisher populations do not exist, maybe a structured model will essentially always fit the data better than a nested unstructured model, especially given that the authors do not seem to use anything like the AIC or BIC penalize the unstructured model for having fewer parameters.

We have now included AIC values in places where we make likelihood comparisons. The AIC for the structured is substantially lower. This is detailed in the Methods section under the subheading “Model convergence and fitting”.

I would not normally think it was reasonable to suggest that a paper needs to be expanded to include analyses of other species, but in this case I think an analysis of at least one or two other species is needed to inform interpretation of the human data. If cobraa tends to find that ancient population structure is a better fit than panmixia in nonhuman species as well, I think that finding would be important and impactful to publish but would be more of a story about methodology rather than the story about human history being currently presented here. Theoretical population geneticists are always asking whether the ancient humps in PSMC plots should be interpreted literally, and empirical users of demographic inference methods are always uncritically interpreting these trajectories as reflections of events like the last glacial maximum; more general insight into what shapes ancient PSMC trajectories would help resolve these conflicting narratives. If instead this ancient structure appears unique to humans, the absence of similar inferences about other species would give me more confidence that populations A and B really existed as discrete entities and reflect something distinctive about human history in particular.

We acknowledge the comments about the tendency of SMC models to generate humps, and the desire to see a broader spectrum of applications to real data sets. We now include analysis with cobraa of a variety of other species (chimpanzee, gorilla, bat, dolphin) and discussion of the variation in results across species. Cobraa gives different results for different species, and indeed finds no ancestral structure in one of the four species, supporting that it does not invariably find some pattern in real data.

It would also be informative to run cobraa on one or more of the high coverage archaic hominid genomes because of this paper’s implicit hypothesis that we should not see similar population structure in their histories (or at least that the contribution of population B should be lower than is seen in humans). One thing I noticed about the PSMC plot in the Pruefer, et al. Altai Neanderthal paper is that the Altai Neanderthal population size trajectory appears to line up with the human population size trajectory during the time period when A and B appear to diverge, which is not what we would expect if Neanderthals were entirely an offshoot of population A.

We spent a considerable amount of time trying to meaningfully run our analysis on the high coverage Neanderthal and Denisovan genomes. The first step of this would be to obtain confident PSMC plots, as published in Pruefer et al. 2014 and Meyer et al. 2012. We regret that we were not able to do this - the right hand side of the plot explodes and is clearly not well constrained. This part of the plot needs to be properly behaved for the subsequent cobraa model to give meaningful results. We have been in contact with Heng Li who generated these original plots, but despite various alterations to the filtering and other elements of the pipeline this could not be resolved. Therefore regrettably we are not able to include analysis of Neanderthal and Denisovan. We now refer to this in the Discussion of our manuscript, and briefly comment further in the methods section when discussing our use of archaic data.

The aspect of this paper that I find most compelling is the fact that the authors are able to classify specific pieces of genomic material as candidate tracts of introgression from “population B.” This posterior decoding opens up many more avenues for sanity checking the authors’ hypothesis than is possible in other studies that have used features of one- or two-locus site frequency spectra to argue for the existence of ancient population structure in Africa. This decoding allows the authors to test regions of A and B ancestry for divergence to Neanderthals and Denisovans, and they find that Neanderthals and Denisovans appear more related to segments of [A] ancestry. However, one thing that gave me pause about this analysis is the numerical range of the probabilities $P(c = BB | t > T_1)$: it looks like there are very few regions where $P(c = BB | t > T_1)$ is higher than 25% ...

... and archaic/human divergence seems to level off starting at $P(c = BB | t > T_1) = 0.05$. If there are 5 times more B/B tracts in the set of regions with 25% posterior probability compared to the set of regions with 5% posterior probability, shouldn’t the set of regions with 25% posterior probability look significantly more diverged from Neanderthals compared to the regions with 5% posterior probability? In contrast, $P(c = AA | t > T_1)$ seems to span the full range from 0 to 1, with lots of regions where $P(c = AA | t > T_1)$ is near 0%. Can the authors offer some insight into why these distributions are so different?

These are reasonable points, and led us to realise that what we were plotting was not in fact what actually matters for the mean divergence from human to archaic. It is the expected fraction of the region that is B that matters, not the fraction of BB or AA. Below we show the originally submitted Figure 5 plots above corresponding plots generated from simulations according to the best fit model for the ESN individual (West African), which have then been decoded using cobraa-path as in Figure 5. The pale blue lines show the fraction of B lineage for genomic locations in each bin. As you can see, these reflect well the observed divergence plots (remarkably well - we were slightly amazed when we did this!), consistent with the suggestion that the A lineage is approximately 0.0012 divergent from the archaics and the B lineage approximately 0.0028 divergent. We agree that the one on the right for BB is less naturally intuitive than that on the left for AA. We expect the reason for this behaviour is that the prior of path BB is only 0.04 given that $\gamma = 0.2$, so estimation of BB is not a good predictor of estimation of AB (prior 0.32) which dominates the overall estimation of the B component.

Figure 5: Relationship between the probability of a present day human region deriving from population A

I am worried that if the model is so biased toward classifying regions as “AA” rather than “BB”, there might be many systematic differences between AA and BB regions that affect downstream analyses. For example, maybe regions with recent coalescence times close to T_1 are more likely to [be] confidently classified as AA and are also more likely to be near genes compared to regions with much older coalescence times. When you simulate data from a history like the one inferred here, do you get such different distributions of AA and BB posterior probabilities?

Yes, when we simulate from the inferred history and apply the model as requested, we obtain similar distributions, as shown above.

Investigating this further, as shown below most of the density is in the region where the estimated $P(c=AA|t>T_1)$ is large, or $p(c=AB|t>T_1)$ is large (for $t>T_1$ there will also be a spike at 0 corresponding to the fraction of time that $T_1<t<T_2$). There are no regions where $p(c=BB|t>T_1)$ is large. This is the same as saying that we don't have power to confidently find BB regions. We have added these plots to the supplement together with the corresponding plots from the simulation, which show that the results are behaving as expected were the model to be true. We would not call this bias - it is correct Bayesian inference given the priors on the different categories and the data available.

Given the non-intuitiveness of the original Figure 5 plots, as revealed by the referee's comments and the investigation above, we have replaced these with alternative plots that are much more direct, as copied here (Neanderthal to the left, Denisovan to the right).

These show dots for 10kb windows spaced every 10Mb. The regions where we assign higher expected A fraction are less divergent. The Spearman correlations are between 0.4 and 0.53, with p-values below $1e-10$. The left hand set of points is around 0.5 on the x-axis because we have low power to identify the BB state.

Figure S14 shows some differences among human populations and archaic genome divergence that seem potentially important but don't appear to be square with the narrative about Neanderthals and Denisovans branching off from population A. This population variation should be highlighted in the main text and discussed in relation to the model the paper is putting forward. The first feature of S14 that jumped out at me is the qualitative difference between $P(c = BB \mid T_{-1} < t < T_{-2})$ and $P(c = BB \mid t > T_{-2})$ —assuming that the red lines are Asian populations, it looks like regions with very old coalescence times and a very low probability of BB ancestry are deeply diverged from Neanderthal/Denisova in East Asians, close to Neanderthal/Denisova in Africans and Amerindians, and intermediately diverged in Europeans? Then there's an opposite and less dramatic ancestry gradient when you look at regions with relatively high probability of BB ancestry, where divergence to Neanderthal/Denisova is lowest for East Asians and highest for Africans. In contrast, variation among populations is low when you look at the relationship between archaic divergence and the posterior probability of AA ancestry. This seems like the opposite of what you'd expect if Neanderthals and Denisovans branched off from population A: in Europeans and East Asians, AA ancestry should on average have more archaic affinity than AA ancestry in Africans, since some of the "A" ancestry in non-African populations should be tracts of archaic introgression. In contrast, "B" ancestry in all populations should be equally diverged from Neanderthals and Denisovans. Maybe my interpretations are off base, but I think these non-intuitive population differences need to be highlighted in the main text and given some explanation.

Given that we have replaced these plots, this discussion is no longer relevant. We believe that the odd behaviours in these rather derived plots for non-Africans are caused by subsequent admixture from archaics after leaving Africa, and further demographic bottlenecks and admixture events that are not included in our model and not relevant to whether the archaics split (primarily) from the A or B lineage. To address this question we believe it is appropriate to focus on African populations which are not subject to these issues. We do this, giving the reason, in the revised text.

Minor comments:

Lines 36-37 say, “for any panmictic model...there necessarily exists a structured model that can generate exactly the same coalescence rate profile.” Isn’t this trivially true? Did you mean to say the converse, that for every structured model there’s a panmictic variable size model with the same rate profile? Or is the idea that you can recapitulate the rate with structure without having to introduce population size changes?

We meant to say that we can recapitulate the rate with structure without having to introduce population size changes. We clarified the text.

The legend of Figure 2 is hard to parse because alpha and beta aren’t defined. I think they’re supposed to be the population size change times but if so that should be annotated in panel a.

We edited the caption to include “ $Q(\alpha|\beta)$ describes the probability of transitioning from discretised time β to time α , conditional on a recombination having occurred.”

At the start of section 3.2, the specification of the structured hypothesis isn’t very well explained. Is the idea to specify a history containing one period of structure with flexible start and end times and allow the model to find T_1 and T_2 in a completely unsupervised way, or is there something encoded in the model that encourages T_1 and T_2 to occur near the start and end points of the ancient population size hump?

Cobraa performs an unconstrained grid search over the values of T_1 and T_2 . There is nothing encoded in the model space to encourage T_1 and T_2 to occur at any specific position. Their optimal locations are driven by the data. For the human data T_1 ends up near “the hump” but T_2 is much more ancient.

However, we realise that the start of section 3.2 was a little confusing. We have moved the first sentence to the end of the penultimate paragraph of the previous section, where we believe it sits better because it concerns demonstration of sufficient information, rather than describing experiments using cobraa itself. And we have consequently reworded the start of section 3.2. We thank the reviewer for prompting these changes.

Reviewer #2

The authors introduce a novel method that can detect structure in the populations ancestral to contemporary individuals. Specifically, the method is an extension of the widely used method PSMC that infers past changes in effective population size from a single diploid genome of a contemporary individual. It has been noted in the literature that different models of structure in the ancestral population can lead to the same profile of inferred ancestral population sizes, when assuming a single panmictic ancestral population. The authors thus extend the framework of PSMC to incorporate structure in the ancestral population, in particular, a model of a single population that splits into two, which subsequently admix at given fractions to form the main lineage of modern human populations.

The authors apply their method to individuals from different populations from the 1000G dataset and the HGDP dataset, and infer that the most likely model is a deep split of an ancestral population ~1.5 million years ago, and the two ancestral populations subsequently admix around 300k years ago, with a ratio of 80%:20%. In addition, the authors augment the hidden state of their model to identify the ancestral population that a particular genomic region in a contemporary individual derives from. Upon investigating region-specific divergence of modern humans and Neanderthals, the authors provide evidence that the major ancestral population (80%) was likely the ancestor of Neanderthals. The authors also perform a GO analysis on the identified regions to investigate their functional importance. In addition, the authors provide evidence that the minor ancestry was selected against.

The manuscript is a timely contribution to efforts aimed at understanding structure in populations ancestral to modern humans, particularly in the context of ghost-populations that left no contemporary descendants, and how this is related to known archaic populations like Neanderthals and Denisovans. The manuscript is well written, the premise clearly explained, the description of the method easy to follow, and the thoughtful analysis provides clear evidence to support the conclusions. I do not have any major points of criticism. However, I have a couple of comments that I think should be addressed to further strengthen the manuscript, and make it accessible to a wider audience.

In particular:

- When reading only the main text, some notation is not well introduced. Some examples include:

- p.4, l.77: The authors write "we now define...", " but the definition is in the supplement. Please introduce Q^S and Q^U conceptually here.

We have done this.

- p. 7, l.140: L_S and L_U are not defined. Please define.

We have done this.

- p.3, l.61: Add: "In Section 9.3.3 of the Supplement, we show that identifiability issues arise when both the sizes of both ancestral population can change, which justifies our choice to set N_B constant." <-- I think it would be good to mention this here already.

We have done this.

- p.5, l.106: The ratio of μ/r is given, but neither a value of μ or r . Please specify. Similarly in the caption of Figure 3. And p.5, l.116.

We have done this.

- Figure 3: In (a), the simulated T_1 and T_2 match the values later inferred for human data approximately. However, for (b), T_1 and T_2 are substantially lower. Was this choice to better exhibit the differences visually? Please provide a simulation like 3(b) where T_1 and

T₂ match the values inferred from real data (approximately). In addition, also provide a simulation study where you simulate data using the N_A(t) (and other parameter) inferred from the real data (and matching mu and r), and present the estimates. This would indicate how robust the inference from the real data is. If there is a reason either of those cannot be provided, please state the reason.

We have remade Figure 3 with new simulations, and also discuss more simulations in the Supplement. As requested, we also have done simulations from the model exactly as inferred in real data, then ran cobraa on this to show that we can infer back the parameters. This is shown in Supplementary Figure 5b and 5c.

- Figure 3(d): Please indicate ticks and the range of values on the x- and y-axes. Does the white-ish square indicate the true value? If so, please indicate more clearly, perhaps a stronger color contrast and bolder outline. Also, is there a way to better the scale of the colorbar then putting -1.6e6 at the top?

We have added the ticks and improved the indication of the true value. We changed the scale to indicate the log-likelihood difference relative to the best fitting unstructured model.

- I was somewhat surprised that in the model selection simulations displayed in Figure 3(e), the difference in log-likelihood for the unstructured simulations can be negative. The authors claim (and I think it is true) that the structured model is nested in the unstructured model. In that case, shouldn't the likelihood difference between the true maxima always be non-negative? The method certainly requires discretization, numerical optimization, and the EM algorithm is only guaranteed to find local optima. Are these the only factors that make the difference negative? Is there some subtlety that I am missing here? Is a difference of -100 cause for concern that the numerical optimization might not identify a "good" optimum? Could you please add a short discussion of these issues?

We believe this is due to the EM algorithm getting stuck in non global optimum. We have added a new simulation section "Inference from cobraa on simulations" in the Supplement that discusses this.

- Somewhat related: Figure 4(c) shows that the log-likelihood difference for the real data is 500 or more. The structured model has more parameters than the unstructured model, so some positive difference is expected. I do not believe that the expected difference will be anywhere near 500 (especially since N_B is assumed constant), and thus, the results hold. But please add a short discussion of the difference in number of parameters, and why it is no cause for concern.

We have done this, and in particular refer to AIC as requested by reviewer 1. We added this discussion in the Methods section under "Model convergence and fitting".

- Figure 4(b): The authors indicate the panmictic population size inferred by PSMC in grey. When looking at this plot, I was also very curious about the inferred N_B. N_B is indicated in Suppl. Figure 6, but I was wondering if there is a good way to integrate it in Figure 4(b), or maybe even provide it in a separate panel in Figure 4.

We have updated the figure to include the estimated size of population B. The shown value is the mean of the maximum likelihood estimates from each population. Extended Figure 3a now shows the N_B value per population. Extended Figure 3b shows the likelihood profile for N_B : we fixed all the other parameters ($N_A, T_1, T_2, \gamma, \rho$) and calculated the likelihood of the data for various values of N_B .

- p.7, l.154: The authors indicate that they use PSMC to obtain estimates of $N_A(t)$ in the panmictic case. I suspect that they do not literally run the software PSMC, but rather cobraa, with the parameters restricted to resemble PSMC. Could you please indicate explicitly whether you run the software PSMC, or cobraa restricted to mirror PSMC.

Indeed, we used the PSMC algorithm as built into the cobraa codebase. We added a line in the Methods indicating this.

- For some of the functional analysis of the admixed regions, the p-values are very strong. This makes me wonder whether chromosomal linkage is correctly accounted for. Specifically, on p.9, l.211, correlation between dcCDS and probability of admixture is computed. However, the probabilities of admixture are auto-correlated along the genome. Is this accounted for when computing the significance of the correlation? If so, how? Furthermore, on p.9, l.225, the significance of a set of genes to overlap extreme values of admixture is computed. The statistic for extremity $H(x)$ is computed in increments of 1kbp. However, genes can be longer than 1kbp, and thus each gene can "try" multiple times to overlap with an extreme position. I wonder whether a binomial with $p=0.01$ is thus the right null model. Also, $H(x)$ is auto-correlated. I do believe that the signals the authors report are significant, but perhaps the p-values could be inflated? Please clarify how these issues are accounted for, if they are indeed issues.

We thank reviewer 2 for this. Indeed, we had not correctly accounted for linkage. We have now recalculated (using all chromosomes now, not just chromosome 1 as in the preprint), taking instead 1kb measurements every 100kb, which we believe is sufficient to avoid nearby correlation caused by linkage disequilibrium. There were tiny changes in the reported Spearman correlation coefficients, and all p-values are still significant (though not as strong as before).

When testing for enrichment with the AAGs or ASGs, we also realised that a binomial test with $p=0.01$ is not the correct null model. We recalculated by looking at how much of the callable sequence (positions where $H(x)$ is calculated) overlaps a gene (according to the start and stop positions as described by HAVANA). We calculated this as 48%. We thus recalculated with a binomial test using $N=21587$ and $p=0.48$ - this means the numbers of AAGs and ASGs are substantially depleted relative to how many we would expect ($N \cdot p=10362$, but AAGs=680 and ASGs=1287). To sanity check this result, we also drew random positions of callable sequence (positions where $H(x)$ is calculated) and indeed found that ~48% overlap genes. We have changed the text to reflect this.

- p.14, l.366: Please indicate that a detailed derivation can be found in the supplement.

We have done this.

- p.15, l.412: Did you use population-specific B-maps?

No, we used the map as inferred in YRI. We believe that this is not a problem as in the paper of Murphy et al they show that the Pearson correlation between different populations' B-maps is always >0.999 .

- p.17, l.465: Please indicate more clearly that the only difference between the analysis in this section and the analysis in the main text is that each chromosome is cut into two "independent" pieces. The nucleotides marked as missing are exactly the same as before. This is a bit hard to follow.

What reviewer 2 writes is not quite correct here, because the nucleotides in the telomeres and centromeres previously marked as missing are now excluded from the analysis entirely. We edited the text to clarify.

- p.18, l.516: How is the divergence calculated? Between only the focal individual and Neanderthal? Or all individuals in the focal population?

Between the focal individual and the Neanderthal (or Denisovan). We now state this in the relevant Methods section.

- Supplementary Figure 4: As for Figure 3(d), please indicate values on the axes and indicate the truth more clearly.

We remade this plot and show instead the relative error.

- Supplementary Figure 15: For some populations, the MLE of T_1 and T_2 is on the edge of the grid used, indicating that the true MLE could lie outside of the grid. This effect would lead to underestimating the true maximum of the CML. I think it would be appropriate to extend the grid such that all individual maxima are truly in the interior. Please do so or comment why this is not necessary.

We have done this, a wider search is now displayed in this Figure. The CML did not change.

- The notation in the derivation of the transitions in Section 9.3 is somewhat inconsistent, which makes it somewhat hard to follow. I think it would be good to streamline this notation for interested readers:

- All derivations assume $t \in [\tau_\alpha, \tau_{\alpha+1}]$ and $s \in [\tau_\beta, \tau_{\beta+1}]$. Please state this clearly, perhaps on p.53, l.946, otherwise it is hard to follow why α appears in section 9.3.1 and subsections.

This definition is given in section (what is now) 10.3.1, in the first paragraph in section "Transition probabilities under the structured mode".

- T_α is never defined to be equal to τ_α . Thus, all instances of T_α and T_β in section 9.3.1 and subsections have to be replaced by τ_α and τ_β . Or the notation made more explicit. The notation T_α is also somewhat ambiguous if T_1 and T_2 exist.

Thanks for noticing this. All instances of T_α have been replaced by τ_α

- The notation λ_{A_α} should be properly introduced.

We have now explicitly introduced this after equation 23 (line 963).

- Many ψ have a subscript β in addition to the case number. This seems obsolete.

The value for ψ in some cases is dependent on where s (continuous version of β) coalesces. E.g. Case 2, ψ_{AA} (equation 36), has a $L_B(T_1, s)$ term in the denominator, so indeed the β is required for some (but not all) ψ .

- Supplementary Figure 27: These figures are a bit hard to interpret: No scale is indicated for the likelihood. Please indicate the scale. Furthermore, the disconnected regions in the likelihood-surface can be an artifact of the discretization grid for the values on the x and y axis. While the likelihood surfaces are shallower in the middle, the difference from one contour line to the next could still be 100 log-likelihood units, which I don't think would be fair to be interpreted as "unidentifiable."

We are very grateful for the acute observations of Reviewer 2. They are correct when they wrote "the disconnected regions in the likelihood-surface can be an artifact of the discretization grid". We have replotted the Figures using a heatmap, which avoids the discretisation artifacts from the contour plot. The general point remains the same that the likelihood surface is difficult to navigate, but we rephrased the text to more accurately describe what the plots show.

Minor points:

- p.3, l.67, "... the coalescence times of the two haploid sequences across the genome."

Done

- p.4, l.81: "Moreover, the relative difference increases when the admixture ... between A and B increases."

Done

- Figure 3(e): Please place a legend in the Figure that indicates the color coding.

Done

- p.5, l.137: Please explicitly indicate that all parameters in this section, except the split times, are as in the previous paragraphs (What is μ/r ?).

Done

- p.8, l.154: "... unstructured (PSMC) (4a) and structured (cobraa) (4b) model ..."

Done

- p.9, l.209: Define CDS.

Done

- p.9, l.220: What does "inferred admixture" mean? Does it mean inferred B ancestry?

Yes, that is what we were using it to mean. We thank reviewer 2 for pointing out that this is misleading, and we have changed the text to "We next looked at regions of the genome that are enriched or depleted of inferred B ancestry".

- Figure 5: The period at the end of subcaption (a) is weirdly spaced and bold. In subcaption (b), "increases" has a bold i.

Done

- Supplementary Figure 1, Caption: "Relative difference in transition matrices for different model parameters."

Done

- Supplementary Figure 1, Caption: "For the matrix λ_i , we plot values in the rows that corresponds to ..."

Done

- Supplementary Figure 2, Caption: "(a) Data are the sequence of true simulated coalescent times ..."

Generally we have avoided the word "true"; we have added in "simulated" instead as that is consistent with the rest of the document.

- Supplementary Figure 2, Caption: Third to last line, the S is not a subscript for L_S .

We were trying to say plural of likelihoods. We have replaced " \mathcal{L}_s " with "likelihoods"

- Supplementary Figure 14: Please indicate the values on the y-axes.

We have done this.

- p.49, l.835: Subscript missing for x: " $x_l = 0$ ", " $x_l = 1$ ", and " $x_l = -1$ ".

We believe reviewer 2 means previous line 833. We have added the subscript.

- p.50, l.869: Doesn't Cardin and McVean's model allow 1 and 3, but not 2?

Yes, absolutely. We thank reviewer 2 for spotting our mistake.

- p.51, l.891: "... in MSMC2 [99] Malaspinas et al. integrate ..."

We have changed the text as requested.

- p.51, l.893: "Similarly, to create the transition matrix ..., we follow Malaspinas et al. and take ..."

We have changed the text as requested.

- p.52, l.917: disallowed

We have changed the text as requested.

- p.54, Equation (21): C^{AB} in numerator.

We have changed the text as requested.

- p.55, Equation (22): In the last line, I think the γ in front of the parentheses is wrong.

We thank reviewer 2 for their sharp eyes and spotting the mistake. We have changed the text as requested - the code was correct and therefore the results are unchanged.

- p.55, Equation (23): Replace T_{j+1} with τ_{j+1} (Same in (24) and (25))

We have changed the text as requested.

- p.55, l.965: " $S \in \{1, \dots, D-1\}$ "

We have changed the text as requested.

- p.55, Equation (26): Last line: T_S is incorrect. Replace λ_{A_b} by λ_{A_α} . Replace t_2 by t . Replace T_b by T_α .

We have changed the text as requested.

- p.56, Equation (27): Replace T_S with T_1 .

We have changed the text as requested.

- p.56, l.971: "Similar to Malaspinas et al. in [99] and Schiffels and Durbin in [38], we use the expected coalescent time $\langle \dots \rangle$ in interval β (see ...)." <-- No integration is performed here.

We have changed the text as requested. The integration being performed is in the referenced equation (11).

- p.56, Equation (30) and (31): I believe there has to be $1/\lambda$ before the inner parentheses (in addition to the one before the outer parentheses).

We thank reviewer 2 for their sharp eyes and spotting the mistake. We have changed the text as requested - the code was correct and therefore the results are unchanged.

- p.56, l.984: The notion of a "solid lineage" is never introduced.

This terminology was introduced in a footnote on page 50 - we thank reviewer 2 for pointing out that this is unclear. We have moved it to the main text.

-p. 57, Equation (36) and (37): In all denominators, t has to be replaced by s . These ψ are also functions of s , so please indicate that.

We thank reviewer 2 for correctly pointing out that in eqs 36 and 37 t needs to be replaced by s . The ψ are indeed functions of s , but β is the discrete form of s and the ψ terms already have a β subscript.

- p.58, Equation (39): Replace T_S with T_1 .

We have changed the text as requested.

- p.58, Equation (40): Some of the "L" have a dash-through.

We believe this is a rendering error for reviewer 2, as it does not appear in any documents when we view them

- p.57, Equation (48) and (49): In all denominators, λ has to be replaced by $\lambda(s)$. These ψ are also functions of s , so please indicate that.

Indeed reviewer two is correct that λ is not written correctly: there is an α subscript which should be β (where β is the discrete form of s). The subscript notation is consistent throughout the document. The ψ are indeed functions of s , but β is the discrete form of s and the ψ terms already have a β subscript.

- p.67, l.1116: The number of parameters has to use the index of T_1 and T_2 , not the times directly.

We have changed the text to accommodate this: we introduced S as the index for T_1 and E as the index for T_2 .

Cobraa paper: Response to reviewers #2

Below we include the reviewers' comments in black and our responses in blue.

Reviewer #1

The authors have done a commendable job with their thoughtful and extensive revisions. I was surprised that this expert team ran into problems trying to reproduce prior PSMC results on archaic genomes, but the paper's other new analyses, particularly the new analyses of non-human genomes, have largely alleviated my skepticism about the authenticity and interpretability of the ancient structure that is being found. The nonhuman results are fascinating in their own right and seem likely to inspire useful new studies of the ubiquity of admixture in nature.

Reviewer #2

The authors have addressed most of my concerns satisfactorily. However, I do think that some minor points remain to be addressed, which I will list below (Page and lines refer to new revised paper). Addressing these points does not affect any of the major results and should be fairly quick. I thus do not think that these necessitate another round of review and could perhaps just be checked by the editor once implemented (and given that there is no major disagreement).

- p.16, l.455: I believe that the enrichment analysis for AAG and ASG vs H(x) is still not entirely correct. The authors claim correctly that ~~40,632~~ 10,362 out of 21,587 1%-H(x) positions (48%) should overlap with genes. But the authors then incorrectly use 680 AAGs or 1287 ASGs to compute the p-value of the enrichment. The value that should be used instead of the number of genes is the number of positions that overlap with these genes, which, judging from Supplementary Figure 8 are likely more than 1 for each gene. Thus the 15-fold enrichment ~~depletion~~ and the p-value of $< 1e-100$ is inflated.

We are very grateful for the diligence of reviewer 2. They are correct that there are many top/bottom H(x) regions that "hit" the same gene.

For the top 1% of H(x), the total number of times the 680 AAGs are hit is 10,991. This is a 1.06-fold enrichment relative to a neutral expectation, which is however still significant with p-value=1.08e-17, according to a binomial test with N=21,587 and p=0.48 (expected = $N * p = 10,362$).

For the bottom 1% of H(x), the total number of times the 1287 ASGs are hit is 19,974. This is a 1.92-fold enrichment, which is strongly significant with p-value $< 1e-100$, according to a binomial test with N=21,587 and p=0.48.

We have changed the text to incorporate this corrected analysis,

We note that in Supplementary Figure 8 we explicitly plot only one line per gene. For example, the two nearby blue lines close to 15Mb are for different genes: KAZN (H(x) position 15,103,000) and TMEM51 (H(x) position 15,185,000).

- Figure 3: The labels for the panels are out of order.

This was deliberate, in order to fit the figure and its caption on one page. If the editorial staff prefer an alternate arrangement of the panels we are happy to oblige.

- p.10, l.243: In the main text, the authors state that PANTHER is used for the GO analysis, whereas in the Methods (p.16, l.466) the authors indicate that geneontology.org is used. Please clarify which program is used for the GO analysis.

geneontology.org is powered by PANTHER. We put the genes into geneontology.org which then used PANTHER for analysis. The geneontology.org website has a page explaining how to cite it <https://geneontology.org/docs/go-citation-policy/>, saying to cite Ashburner et al. (2000) and the Gene Ontology consortium (2023) for GO and the Thomas et al. (2022) PANTHER paper for GO enrichment analysis, which is exactly what we do.

- Section 9 "Supplementary Text" on p.36 is empty.

We have amended this, so that now there is a Section 9 "Supplementary Figures and Tables", and afterwards another Section 10 "Supplementary Text".

- p.60, l.1095: Remove the duplicate "and".

We have done this.

- p.60, l.1096: s is not integrated out, it is set to the mean value in the respective interval. Please correct.

We thank reviewer 2 for their sharp eyes. We have changed the text from
"we use the expected coalescence time $\langle t_{\beta} \rangle$ in interval β to integrate out s "

To

"we use the expected coalescence time $\langle t_{\beta} \rangle$ in interval β to discretise s "

- p.61, Equation (35), (36), p.64, Equation (47), (48): Replace s on the right-hand side by $\langle t_{\beta} \rangle$.

We thank reviewer 2 for their sharp eyes. Indeed we have changed equations (35), (36), (47), and (48). However, to be more consistent with the rest of the document, instead of writing $\lambda_A(\langle t_{\beta} \rangle)$, we wrote $\lambda_{A_{\beta}}$. We introduced this notation on line 1091 and 1092 (numbering as in the newly submitted version), and it is employed widely in the document already (the first time it appears is in equation (25), it is then seen in (26), (26-30), and very often thereafter.

- p.16, l.449: Please indicate in the manuscript that the YRI B-maps were used, and that this is reasonable since the correlation between different population B-maps is very high.

In the Methods we added the sentence “We believe that using the YRI B-map for all populations is sufficient, because Murphy et al. report that B-maps inferred in different populations all have a Pearson correlation of >0.999 with the YRI B-map”. This comes from their Appendix 1 - Figure 35.

- Supplementary Figure 4: Please add ticks on the x- and y-axis to the figure.

We have done this.

- p. 78, l. 1335: I do not agree with the authors that Supplementary Figure 24 provides evidence for multiple local maxima. Yes, a lot of cells are yellow-ish, but that makes it just impossible to deduce from the plots how many maxima there are. I think that statement should be rephrased a bit more carefully.

We have edited the text to reflect this. We now say “Figure S24a illustrates that there is an extended region in the space of ($\hat{\lambda}_{A_{10:18}}$, $\hat{\lambda}_{B_{10:18}}$) values that achieves a likelihood comparable to the simulated value, seemingly by trading off changes in the two variables.” We also edited the following paragraph slightly to similarly reflect what can be seen from the plots.